# Locality-Aware
# Generalizable Implicit Neural Representation

**Doyup Lee**[*]
Kakao Brain
doyup.lee@kakaobrain.com

**Chiheon Kim**[*]
Kakao Brain
chiheon.kim@kakaobrain.com

**Minsu Cho** [†]
POSTECH
mscho@postech.ac.kr

**Wook-Shin Han** [†]
POSTECH
wshan@dblab.postech.ac.kr

## Abstract

Generalizable implicit neural representation (INR) enables a single continuous function, i.e., a coordinate-based neural network, to represent multiple data instances by modulating its weights or intermediate features using latent codes. However, the expressive power of the state-of-the-art modulation is limited due to its inability to localize and capture fine-grained details of data entities such as specific pixels and rays. To address this issue, we propose a novel framework for generalizable INR that combines a transformer encoder with a locality-aware INR decoder. The transformer encoder predicts a set of latent tokens from a data instance to encode local information into each latent token. The locality-aware INR decoder extracts a modulation vector by selectively aggregating the latent tokens via cross-attention for a coordinate input and then predicts the output by progressively decoding with coarse-to-fine modulation through multiple frequency bandwidths. The selective token aggregation and the multi-band feature modulation enable us to learn locality-aware representation in spatial and spectral aspects, respectively. Our framework significantly outperforms previous generalizable INRs and validates the usefulness of the locality-aware latents for downstream tasks such as image generation.

## 1 Introduction

Recent advances in generalizable implicit neural representation (INR) enable a single coordinate-based multi-layer perceptron (MLP) to represent multiple data instances as a continuous function. Instead of per-sample training of individual coordinate-based MLPs, generalizable INR extracts latent codes of data instances [13, 14, 40] to modulate the weights or intermediate features of the shared MLP model [8, 11, 19, 35]. However, despite the advances in previous approaches, their performance is still insufficient compared with individual training of INRs per sample.

We postulate that the expressive power of generalizable INRs is limited by the ability of *locality-awareness* to localize relevant entities from a data instance and control

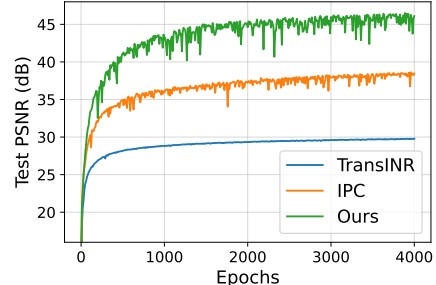

Figure 1: Learning curves of PSNRs during training on ImageNette 178×178.

---

[*]Equal contribution
[†]Corresponding authors

37th Conference on Neural Information Processing Systems (NeurIPS 2023).

their structure in a fine-grained manner. Primitive entities of a data instance, such as pixels in an image, tend to have a higher correlation with each other if they are closer in space and time. Thus, this locality of data entities has been used as an important inductive bias for learning the representations of complex data [3]. However, previous approaches to generalizable INRs are not properly designed to leverage the locality of data entities. For example, when latent codes modulate intermediate features [11, 12] or weight matrices [8, 19, 35] of an INR decoder, the modulation methods do not exploit a specified coordinates for decoding, which restricts the latent codes to encoding global information over all pixels without capturing local relationships between specific pixels.

To address this issue, we propose a novel encoder-decoder framework for *locality-aware* generalizable INR to effectively localize and control the fine-grained details of data. In our framework, a Transformer encoder [37] first extracts locally relevant information from a data instance and predicts a set of latent tokens to encode different local information. Then, our locality-aware INR decoder effectively leverages the latent tokens to predict fine-grained details. Specifically, given an input coordinate, our INR decoder uses a cross-attention to selectively aggregate the local information in the latent tokens and extract a modulation vector for the coordinate. In addition, our INR decoder effectively captures the high-frequency details in the modulation vector by decomposing it into multiple bandwidths of frequency features and then progressively composing the intermediate features. We conduct extensive experiments to demonstrate the high performance and efficacy of our locality-aware generalizable INR on benchmarks as shown in Figure 1. In addition, we show the potential of our locality-aware INR latents to be utilized for downstream tasks such as image synthesis.

Our main contributions can be summarized as follows: 1) We propose an effective framework for generalizable INR with a Transformer encoder and locality-aware INR decoder. 2) The proposed INR decoder with selective token aggregation and multi-band feature modulation can effectively capture the local information to predict the fine-grained data details. 3) The extensive experiments validate the efficacy of our framework and show its applications to a downstream image generation task.

## 2 Related Work

**Implicit neural representations (INRs).** INRs use neural networks to represent complex data such as audio, images, and 3D scenes, as continuous functions. Especially, incorporating Fourier features [24, 36], periodic activations [31], or multi-grid features [25] significantly improves the performance of INRs. Despite its broad applications [1, 6, 10, 32, 34], INRs commonly require separate training of MLPs to represent each data instance. Thus, individual training of INRs per sample does not learn common representations in multiple data instances.

**Generalizable INRs.** Previous approaches focus on two major components for generalizable INRs; latent feature extraction and modulation methods. Auto-decoding [23, 26] computes a latent vector per data instance and concatenates it with the input of a coordinate-based MLP. Given input data, gradient-based meta-learning [4, 11, 12] adapts a shared latent vector using a few update steps to scale and shift the intermediate activations of the MLP. Learned Init [35] also uses gradient-based meta-learning but adapts whole weights of the shared MLP. Although auto-decoding and gradient-based meta-learning are agnostic to the types of data, their training is unstable on complex and large-scale datasets. TransINR [8] employs the Transformer [37] as a hypernetwork to predict latent vectors to modulate the weights of the shared MLP. In addition, Instance Pattern Composers [19] have demonstrated that modulating the weights of the second MLP layer is enough to achieve high performance of generalizable INRs. Our framework also employs the Transformer encoder, but focuses on extracting locality-aware latent features for the high performance of generalizable INR.

**Leveraging Locality of Data for INRs** Local information in data has been utilized for efficient modeling of INRs, since local relationships between data entities are widely used for effective process of complex data [3]. Given an input coordinate, the coordinate-based MLP only uses latent vectors nearby the coordinate, after a CNN encoder extracts a 2D grid feature map of an image for super-resolution [7] and reconstruction [22]. Spatial Functa [4] demonstrates that leveraging the locality of data enables INRs to be utilized for downstream tasks such as image recognition and generation. Local information in 3D coordinates has also been effective for scene modeling as a hybrid approach using 3D feature grids [18] or the part segmentation [17] of a 3D object. However, previous approaches assume explicit grid structures of latents tailored to a specific data type. Since

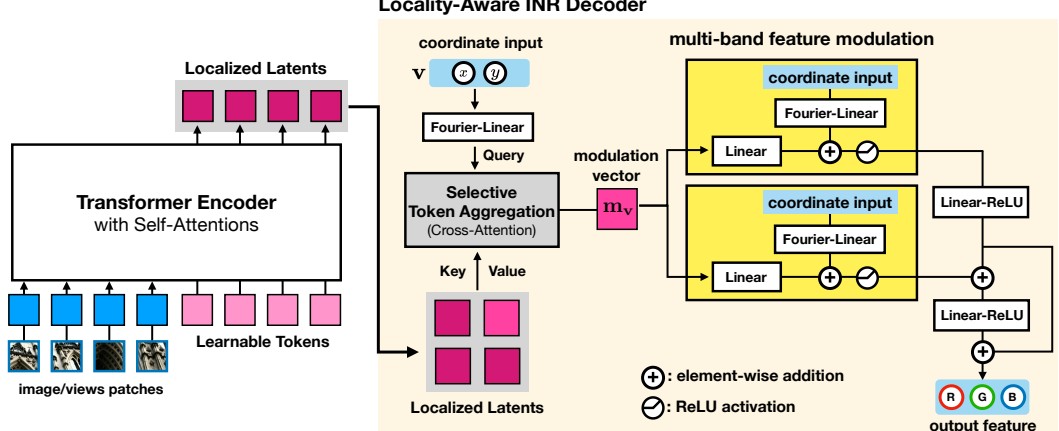

Figure 2: Overview of our framework for locality-aware generalizable INR. Given a data instance, Transformer encoder extracts its localized latents. Then, the locality-aware INR decoder uses selective token aggregation and multi-band feature modulation to predict the output for the input coordinate.

we do not predefine a relationship between latents, our framework is flexible to learn and encode the local information of both grid coordinates in images and non-grid coordinates in light fields.

## 3 Methods

We propose a novel framework for *locality-aware generalizable INR* which consists of a Transformer encoder to localize the information in data into latent tokens and a locality-aware INR decoder to exploit the localized latents and predict outputs. First, we formulate how generalizable INR enables a single coordinate-based neural network to represent multiple data instances as a continuous function by modulating its weights or features. Then, after we introduce the Transformer encoder to extract a set of latent tokens from input data instances, we explain the details of the locality-aware INR decoder, where *selective token selection* aggregates the spatially local information for an input coordinate via cross-attention; *multi-band feature modulation* leverages a different range of frequency bandwidths to progressively decode the local information using coarse-to-fine modulation in the spectral domain.

### 3.1 Generalizable Implicit Neural Representation

Given a set of data instances $\mathcal{X} = \{\mathbf{x}^{(n)}\}_{n=1}^N$, each data instance $\mathbf{x}^{(n)} = \{(\mathbf{v}_i^{(n)}, \mathbf{y}_i^{(n)})\}_{i=1}^{M_n}$ comprises $M_n$ pairs of an input coordinate $\mathbf{v}_i^{(n)} \in \mathbb{R}^{d_{\text{in}}}$ and the corresponding output feature $\mathbf{y}_i^{(n)} \in \mathbb{R}^{d_{\text{out}}}$. Conventional approaches [24, 31, 36] adopt individual coordinate-based MLPs to train and memorize each data instance $\mathbf{x}^{(n)}$. Thus, the coordinate-based MLP cannot be reused and generalized to represent other data instances, requiring per-sample optimization of MLPs for unseen data instances.

A generalizable INR uses a single coordinate-based MLP as a shared INR decoder $F_\theta : \mathbb{R}^{d_{\text{in}}} \to \mathbb{R}^{d_{\text{out}}}$ to represent multiple data instances as a continuous function. Generalizable INR [8, 11, 12, 19, 26] extracts the $R$ number of latent codes $\mathbf{Z}^{(n)} = \{\mathbf{z}_k^{(n)} \in \mathbb{R}^d\}_{k=1}^R$ from a data instance $\mathbf{x}^{(n)}$. Then, the latents are used for the INR decoder to represent a data instance $\mathbf{x}^{(n)}$ as $\mathbf{y}_i^{(n)} = F_\theta(\mathbf{v}_i^{(n)}; \mathbf{Z}^{(n)})$, while updating the parameters $\theta$ and latents $\mathbf{Z}^{(n)}$ to minimize the errors over $\mathcal{X}$:

$$\min_{\theta, \mathbf{Z}^{(n)}} \frac{1}{N M_n} \sum_{n=1}^N \sum_{i=1}^{M_n} \left\| \mathbf{y}_i^{(n)} - F_\theta(\mathbf{v}_i^{(n)}; \mathbf{Z}^{(n)}) \right\|_2^2. \tag{1}$$

We remark that each previous approach employs a different number of latent codes to modulate a coordinate-based MLP. For example, a single latent vector ($R = 1$) is commonly extracted to modulate intermediate features of the MLP [11, 12, 26], while a multitude of latents ($R > 1$) are used to modulate its weights [8, 19, 35]. While we modulate the features of MLP, we extract a set of latent codes to localize the information of data to leverage the locality-awareness for latent features.

## 3.2 Transformer Encoder

Our framework employs a Transformer encoder [37] to extract a set of latents $\mathbf{Z}^{(n)}$ for each data instance $\mathbf{x}^{(n)}$ as shown in Figure 2. After a data instance, such as an image or multi-view images, is patchified into a sequence of data tokens, we concatenate the patchified tokens into a sequence of $R$ learnable tokens as the encoder input. Then, the Transformer encoder extracts a set of latent tokens, where each latent token corresponds to an input learnable token. Note that the permutation-equivariance of self-attention in the Transformer encoder enables us not to predefine the local structure of data and the ordering of latent tokens. During training, each latent token learns to capture the local information of data, while covering whole regions to represent a data instance. Thus, whether a data instance is represented on a grid or non-grid coordinate, our framework is flexible to encode various types of data into latent tokens, while learning the local relationships of latent tokens during training.

## 3.3 Locality-Aware Decoder for Implicit Neural Representations

We propose the locality-aware INR decoder in Figure 2 to leverage the local information of data for effective generalizable INR. Our INR decoder comprises two primary components: i) *Selective token aggregation via cross attention* extracts a modulation vector for an input coordinate to aggregate spatially local information from latent tokens. ii) *Multi-band feature modulation* decomposes the modulation vector into multiple bandwidths of frequency features to amplify the high-frequency features and effectively predict the details of outputs.

### 3.3.1 Selective Token Aggregation via Cross-Attention

We remark that encoding locality-aware latent tokens is not straightforward since the self-attentions in Transformer do not guarantee a specific relationship between tokens. Thus, the properties of the latent tokens are determined by a modulation method for generalizable INR to exploit the extracted latents. For example, given an input coordinate $\mathbf{v}$ and latent tokens $\{\mathbf{z}_1, ..., \mathbf{z}_R\}$, a straightforward method can use Instance Pattern Composers [19] to construct a modulation weight $\mathbf{W}_{\mathrm{m}} = [\mathbf{z}_1, ..., \mathbf{z}_R]^\top \in \mathbb{R}^{R \times d_{\mathrm{in}}}$ and extract a modulation vector $\mathbf{m}_{\mathbf{v}} = \mathbf{W}_{\mathrm{m}} \mathbf{v} = [\mathbf{z}_1^\top \mathbf{v}, ..., \mathbf{z}_R^\top \mathbf{v}]^\top \in \mathbb{R}^R$. However, the latent tokens cannot encode the local information of data, since each latent token equally influences each channel of the modulation vector regardless of the coordinate locations (see Section 4.3).

Our selective token aggregation employs cross-attention to aggregate the spatially local latents nearby the input coordinate, while guiding the latents to be locality-aware. Given a set of latent tokens $\mathbf{Z}^{(n)} = \{\mathbf{z}_k^{(n)}\}_{k=1}^R$ and a coordinate $\mathbf{v}_i^{(n)}$, a modulation feature vector $\mathbf{m}_{\mathbf{v}_i}^{(n)} \in \mathbb{R}^d$ shifts the intermediate features of an INR decoder to predict the output, where $d$ is the dimensionality of hidden layers in the INR decoder. For the brevity of notation, we omit the superscript $n$ and subscript $i$.

**Frequency features** We first transform an input coordinate $\mathbf{v} = (v_1, \cdots, v_{d_{\mathrm{in}}}) \in \mathbb{R}^{d_{\mathrm{in}}}$ into frequency features using sinusoidal positional encoding [31, 36]. We define the Fourier features $\gamma_\sigma(\mathbf{v}) \in \mathbb{R}^{d_{\mathrm{F}}}$ with bandwidth $\sigma > 1$ and feature dimensionality $d_{\mathrm{F}}$ as

$$\gamma_\sigma(\mathbf{v}) = [\cos(\pi \omega_j v_i), \sin(\pi \omega_j v_i) : i = 1, \cdots, d_{\mathrm{in}}, j = 0, \cdots, n-1] \tag{2}$$

where $n = \frac{d_{\mathrm{F}}}{2 d_{\mathrm{in}}}$. A frequency $\omega_j = \sigma^{j/(n-1)}$ is evenly distributed between 1 and $\sigma$ on a log-scale. Based on the Fourier features, we define the *frequency feature* extraction $\mathbf{h}_{\mathrm{F}}(\cdot)$ as

$$\mathbf{h}_{\mathrm{F}}(\mathbf{v}; \sigma, \mathbf{W}, \mathbf{b}) = \mathrm{ReLU}\left(\mathbf{W}\gamma_\sigma(\mathbf{v}) + \mathbf{b}\right), \tag{3}$$

where $\mathbf{W} \in \mathbb{R}^{d \times d_{\mathrm{F}}}$ and $\mathbf{b} \in \mathbb{R}^d$ are trainable parameters for frequency features, $d$ denotes the dimensionality of hidden layers in the INR decoder.

**Selective token selection via cross-attention** To predict corresponding output $\mathbf{y}$ to the coordinate $\mathbf{v}$, we adopt a cross-attention to extract a modulation feature vector $\mathbf{m}_{\mathbf{v}} \in \mathbb{R}^d$ based on the latent tokens $\mathbf{Z} = \{\mathbf{z}_k\}_{k=1}^R$. We first extract the frequency features of the coordinate $\mathbf{v}$ in Eq (3) as the query of the cross-attention as

$$\mathbf{q}_{\mathbf{v}} := \mathbf{h}_{\mathrm{F}}(\mathbf{v}; \sigma_{\mathrm{q}}, \mathbf{W}_{\mathrm{q}}, \mathbf{b}_{\mathrm{q}}), \tag{4}$$

where $\mathbf{W}_{\mathrm{q}} \in \mathbb{R}^{d \times d_{\mathrm{F}}}$ and $\mathbf{b}_{\mathrm{q}} \in \mathbb{R}^d$ are trainable parameters, and $\sigma_{\mathrm{q}}$ is the bandwidth for query frequency features. The cross-attention in Figure 2 enables the query to select latent tokens, aggregate

its local information, and extract the modulation feature vector $\mathbf{m_v}$ for the input coordinate:

$$\mathbf{m_v} := \text{MultiHeadAttention}(\text{Query} = \mathbf{q_v}, \text{Key} = \mathbf{Z}, \text{Value} = \mathbf{Z}). \tag{5}$$

An intuitive implementation for selective token aggregation can employ hard attention to select only one latent token for each coordinate. However, in our primitive experiment, using hard attention leads to unstable training and a latent collapse problem that selects only few latent tokens. Meanwhile, multi-head attentions encourage each latent token to easily learn the locality in data instances.

### 3.3.2 Multi-Band Feature Modulation in the Spectral Domain

After the selective token aggregation extracts a modulation vector $\mathbf{m_v}$, we use multi-band feature modulation to effectively predict the details of outputs. Although Fourier features [24, 36] reduce the spectral bias [2, 28] of neural networks, adopting a simple stack of MLPs to INRs still suffers from capturing the high-frequency data details. To address this issue, we use a different range of frequency bandwidths to decompose the modulation vector into multiple frequency features in the spectral domain. Then, our multi-band feature modulation uses the multiple frequency features to progressively decode the intermediate features, while encouraging a deeper MLP path to learn higher frequency features. Note that the coarse-to-fine approach in the spectral domain is analogous to the locally hierarchical approach in the spatial domain [21, 29, 39] to capture the data details.

**Extracting multiple modulation features with different frequency bandwidths** We extract $L$ level of modulation features $\mathbf{m_v}^{(1)}, \cdots, \mathbf{m_v}^{(L)}$ from $\mathbf{m_v}$ using different bandwidths of frequency features. Given $L$ frequency bandwidths as $\sigma_1 \geq \sigma_2 \geq \cdots \geq \sigma_L \geq \sigma_\mathrm{q}$, we use Eq (3) to extract the $\ell$-th level of frequency features of an input coordinate $\mathbf{v}$ as

$$(\mathbf{h}_\mathrm{F})_\mathbf{v}^{(\ell)} := \mathbf{h}_\mathrm{F}(\mathbf{v}; \sigma_\ell, \mathbf{W}_\mathrm{F}^{(\ell)}, \mathbf{b}_\mathrm{F}^{(\ell)}) = \text{ReLU}\left(\mathbf{W}_\mathrm{F}^{(\ell)} \gamma_{\sigma_\ell}(\mathbf{v}) + \mathbf{b}_\mathrm{F}^{(\ell)}\right), \tag{6}$$

where $\mathbf{W}_\mathrm{F}^{(\ell)}$ and $\mathbf{b}_\mathrm{F}^{(\ell)}$ are trainable parameters and shared across data instances. Then, the $\ell$-th modulation vector $\mathbf{m_v}^{(\ell)}$ is extracted from the modulation vector $\mathbf{m_v}$ as

$$\mathbf{m_v}^{(\ell)} := \text{ReLU}\left((\mathbf{h}_\mathrm{F})_\mathbf{v}^{(\ell)} + \mathbf{W}_\mathrm{m}^{(\ell)} \mathbf{m_v} + \mathbf{b}_\mathrm{m}^{(\ell)}\right), \tag{7}$$

with a trainable weight $\mathbf{W}_\mathrm{m}^{(\ell)}$ and bias $\mathbf{b}_\mathrm{m}^{(\ell)}$. Considering that $\text{ReLU}$ cutoffs the values below zero, we assume that $\mathbf{m_v}^{(\ell)}$ filters out the information of $\mathbf{m_v}$ based on the $\ell$-th frequency patterns of $(\mathbf{h}_\mathrm{F})_\mathbf{v}^{(\ell)}$.

**Multi-band feature modulation** After decomposing a modulation vector into multiple features with different frequency bandwidths, we progressively compose the $L$ modulation features by applying a stack of nonlinear operations with a linear layer and ReLU activation. Starting with $\mathbf{h_v}^{(1)} = \mathbf{m_v}^{(1)}$, we compute the $\ell$-th hidden features $\mathbf{h_v}^{(\ell)}$ for $\ell = 2, \cdots, L$ as

$$\widetilde{\mathbf{h}}_\mathbf{v}^{(\ell)} := \mathbf{m_v}^{(\ell)} + \mathbf{h_v}^{(\ell-1)} \quad \text{and} \quad \mathbf{h_v}^{(\ell)} := \text{ReLU}(\mathbf{W}^{(\ell)} \widetilde{\mathbf{h}}_\mathbf{v}^{(\ell)} + \mathbf{b}^{(\ell)}), \tag{8}$$

where $\mathbf{W}^{(\ell)} \in \mathbb{R}^{d \times d}$ and $\mathbf{b}^{(\ell)} \in \mathbb{R}^d$ are trainable weights and biases of the INR decoder. $\widetilde{\mathbf{h}}_\mathbf{v}^{(\ell)}$ denotes the $\ell$-th pre-activation of INR decoder for coordinate $\mathbf{v}$. Note that the modulation features with high-frequency bandwidth can be processed by more nonlinear operations than the features with lower frequency bandwidths, considering that high-frequency features contain more complex signals.

Finally, the output $\hat{\mathbf{y}}$ is predicted using all intermediate hidden features of the INR decoder as

$$\hat{\mathbf{y}} := \sum_{\ell=1}^{L} f_\text{out}^{(\ell)}(\mathbf{h_v}^{(\ell)}), \tag{9}$$

where $f_\text{out}^{(\ell)} : \mathbb{R}^d \to \mathbb{R}^{d_\text{out}}$ are a linear projection into the output space. Although utilizing only $\mathbf{h_v}^{(L)}$ is also an option to predict outputs, skip connections of all intermediate features into the output layer enhances the robustness of training to the hyperparameter choices.

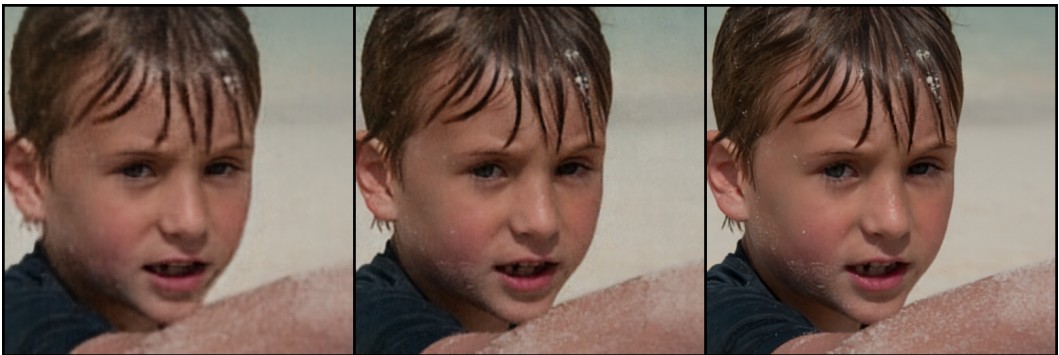

Figure 3: Reconstructed images of FFHQ with 512×512 resolution by TransINR [8] (left), IPC [19] (middle), and our locality-aware generalizable INR (right).

## 4  Experiments

We conduct extensive experiments to demonstrate the effectiveness of our locality-aware generalizable INR on image reconstruction and novel view synthesis. In addition, we conduct in-depth analysis to validate the efficacy of our selective token aggregation and multi-band feature modulation to localize the information of data to capture fined-grained details. We also show that our locality-aware latents can be utilized for image generation by training a generative model on the extracted latents. Our implementation and experimental settings are based on the official codes of Instance Pattern Composers [19] for a fair comparison. We attach the implementation details to Appendix A.

### 4.1  Image Reconstruction

We follow the protocols in previous studies [8, 19, 35] to evaluate our framework on image reconstruction of CelebA, FFHQ, and ImageNette with 178×178 resolution. Our framework also outperforms previous approaches on high-resolution images with 256×256, 512×512, and 1024×1024 resolutions of FFHQ. We compare our framework with Learned Init [35], TransINR [8], and IPC [19]. The Transformer encoder predicts $R = 256$ latent tokens, while the INR decoder uses $d_{in} = 2$, $d_{out} = 3$, $d = 256$ dimensionality of hidden features, $L = 2$, $\sigma_q = 16$ and $(\sigma_1, \sigma_2) = (128, 32)$ bandwidths.

**178×178 Image Reconstruction**  Table 1 shows that our generalizable INR significantly outperforms previous methods by a large margin. We remark that TransINR, IPC, and our framework use the same capacity of the Transformer encoder, latent tokens, and INR decoder except for the modulation methods. Thus, the results imply that our locality-aware INR decoder with selective token aggregation and multi-band feature modulation is effective to capture local information of data and fine-grained details for high-quality image reconstruction.

Table 1: PSNRs of reconstructed images of 178×178 CelebA, FFHQ, and ImageNette.

|  | CelebA | FFHQ | ImageNette |
|---|---|---|---|
| Learned Init [35] | 30.37 | - | 27.07 |
| TransINR | 33.33 | 33.66 | 29.77 |
| IPC | 35.93 | 37.18 | 38.46 |
| Ours | **50.74** | **43.32** | **46.10** |

**High-Resolution Image Reconstruction**  We further evaluate our framework on the reconstruction of FFHQ images with 256×256, 512×512, 1024×1024 resolutions to demonstrate our effectiveness to capture fine-grained data details in Table 2. Although the performance increases as the MLP dimensionality $d$ and the number of latents $R$ increases, we use the same experimental setting with 178×178 im-age reconstruction to validate the efficacy of our framework. Our framework consistently achieves

Table 2: PSNRs on the reconstructed FFHQ with 256×256, 512×512, and 1024×1024 resolutions.

|  | 256×256 | 512×512 | 1024×1024 |
|---|---|---|---|
| TransINR | 30.96 | 29.35 | - |
| IPC [19] | 34.68 | 31.58 | 28.68 |
| Ours | **39.88** | **35.43** | **31.94** |

higher PSNRs than TransINR and IPC for all resolutions. Figure 3 also shows that TransINR and IPC cannot reconstruct the fine-grained details of a 512×512 image, but our framework provides a

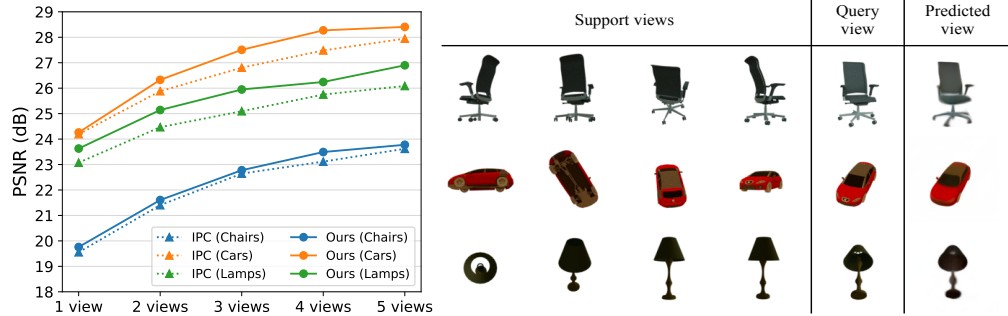

Figure 4: (a) PSNRs on novel view synthesis of ShapeNet Chairs, Cars, and Lamps according to the number of support views (1-5 views). (b) Examples of novel view synthesis with 4 support views.

high-quality result of reconstructed images. The results demonstrate that leveraging the locality of data is crucial for generalizable INR to model complex and high-resolution data.

## 4.2 Few-Shot Novel View Synthesis

We evaluate our framework on novel view synthesis with the ShapeNet Chairs, Cars, and Lamps datasets to predict a rendered image of a 3D object under an unseen view. Given few views of an object with known camera poses, we employ a light field [32] for novel view synthesis. A light field does not use computationally intensive volume rendering [24] but directly predicts RGB colors for the input coordinate for rays with $d_{in} = 6$ using the Plücker coordinate system. Our INR decoder uses $d = 256$ and two levels of feature modulations with $\sigma_q = 2$ and $(\sigma_1, \sigma_2) = (8, 4)$.

Figure 4(a) shows that our framework outperforms IPC for novel view synthesis. Our framework shows competitive performance with IPC when only one support view is provided. However, the performance of our framework is consistently improved as the number of support views increases, while outperforming the results of IPC. Note that defining a local relationship between rays is not straightforward due to its non-grid property of the Plücker coordinate. Our Transformer encoder can learn the local relationship between rays to extract locality-aware latent tokens during training and achieve high performance. We analyze the learned locality of rays encoded in the extracted latents in Section 4.3. Figure 4(b) shows that our framework correctly predicts the colors and shapes of a novel view corresponding to the support views, although the predicted views are blurry due to the lack of training objectives with generative modeling. We expect that combining our framework with generative models [5, 38] to synthesize a photorealistic novel view is an interesting future work.

## 4.3 In-Depth Analysis

**Learning Curves on ImageNette 178×178** Figure 1 juxtaposes the learning curves of our framework and previous approaches on ImageNette 178×178. Note that TransINR, IPC, and our framework use the same Transformer encoder to extract data latents, while adopting different modulation methods. While the training speed of our framework is about 80% of the speed of IPC, we remark our framework achieves the test PSNR of 38.72 after 400 epochs of training, outperforming the PSNR of 38.46 achieved by IPC trained for 4000 epochs, hence resulting in 8× speed-up of training time. That is, our locality-aware latents enables generalizable INR to be both efficient and effective.

**Selective token aggregation and multi-band feature modulations** We conduct an ablation study on image reconstruction of with ImageNette 178×178 and FFHQ 256×256, novel view synthesis with Lamp-3 views to validate the effectiveness of the selective token aggregation and the multi-band feature modulation. We replace the multi-band feature modulations with a simple stack of MLPs (ours w/o multiFM), and the selective token aggregation with the weight modulation of IPC (ours w/o STA). If both two modules are replaced together, the INR decoder becomes the same

Table 3: Ablation study on ImageNette 178×178, FFHQ 256×256, and Lamp-3 views.

|            | ImageNette | FFHQ  | Lamp  |
|------------|------------|-------|-------|
| Ours       | **37.46**  | **38.01** | **26.00** |
| w/o STA    | 34.54      | 34.52 | 25.31 |
| w/o multiFM| 33.90      | 33.65 | 25.78 |
| IPC [19]   | 34.11      | 34.68 | 25.09 |

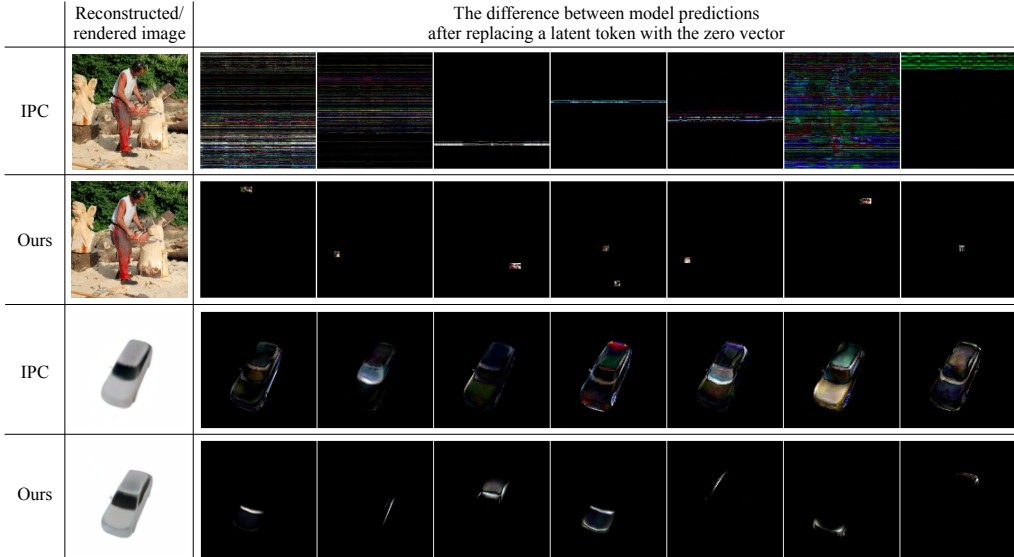

|  | Reconstructed/
rendered image | The difference between model predictions
after replacing a latent token with the zero vector |
|---|---|---|
| IPC | | |
| Ours | | |
| IPC | | |
| Ours | | |

Figure 5: Visualization of differences between model predictions after replacing a latent token with the zero vector, for IPC [19] and our framework.

architectrure as IPC. We use single-head cross-attention for the selective token aggregation to focus on the effect of two modules. Table 3 demonstrates that both the selective token aggregation and the multi-band feature modulation are required for the performance improvement, as there is no significant improvement when only one of the modules is used.

**Choices of frequency bandwidths** Table 4 shows that the ordering of frequency bandwidths in Eq. (4) and Eq. (6) can affect the performance. We train our framework with two-level feature modulations on ImageNette $178 \times 178$ during 400 epochs with different settings of the bandwidths $\sigma_1, \sigma_2, \sigma_q$. Although our framework outperforms IPC regardless of the bandwidth settings, the best PSNR is achieved with $\sigma_1 \geq \sigma_2 \geq \sigma_q$. The results imply that selective token aggregation does not require high-frequency features, but the high-frequency features need to be processed by more nonlinear operations than lower-frequency features as discussed in Section 3.3.2.

Table 4: PSNRs of reconstructed ImageNette $178 \times 178$ with various frequency bandwidths.

| $(\sigma_1, \sigma_2)$ | $\sigma_q$ | ImageNette |
|---|---|---|
| (128, 32) | 16 | **37.46** |
| (32, 128) | 16 | 35.00 |
| (128, 128) | 16 | 35.30 |
| (128, 32) | 128 | 35.58 |
| IPC ($\sigma = 128$) | | 34.11 |

**The role of extracted latent tokens** Figure 5 shows that our framework encodes the local information of data into each latent token, while IPC cannot learn the locality in data coordinates. To visualize the information in each latent token, we randomly select a latent token to be replaced with the zero vector. Then, we visualize the difference between the model predictions with or without the replacement. Each latent token of our framework encapsulates the local information in different regions of images and light fields. However, the latent tokens of IPC cannot exploit the local information of data, while encoding the global information over whole coordinates. Note that our framework *learns* the structure of locality in light fields during training, although the structure of the Plücker coordinate system is not regular as the grid coordinates of images. Thus, our framework can learn the locality-aware latents of data for generalizable INR regardless of the types of coordinate systems.

## 4.4 Generating INRs for Conditional Image Synthesis

We examine the potentials of the extracted latent tokens to be utilized for a downstream task such as class-conditional image generation of ImageNet [9]. Note that we cannot use the architecture of U-Net in conventional image diffusion models [4, 30], since our framework is not tailored to the 2D grid coordinate. Thus, we adopt a Transformer-based diffusion model [15, 27] to predict a set of latent tokens after corrupting the latents by Gaussian noises.

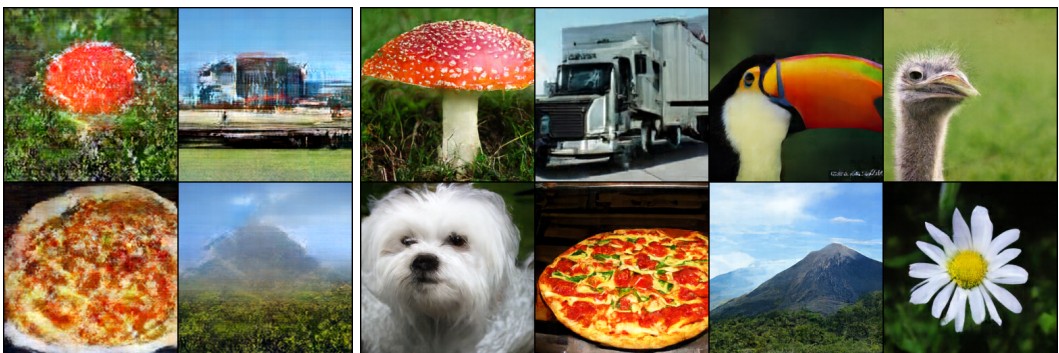

Figure 6: The examples of generated 256×256 images by generating latents of IPC (left) and ours (right), trained on ImageNet.

We train 458M parameters of Transformers during 400 epochs to generate our locality-aware latent tokens. We attach the detailed setting in Appendix A.3. When we train a diffusion model to generate latent tokens of IPC in Figure 6, the generated images suffer from severe artifacts, because the prediction error of each latent token for IPC leads to the artifacts over all coordinates. Contrastively, the diffusion model for our locality-aware latents generates realistic images.

Table 5: Reconstructed PSNRs and FID of generated images on ImageNet 256×256.

|  | Latent Shape | rPSNR | FID |
|---|---|---|---|
| Ours | 256×256 | 37.7 | 9.3 |
| Spatial | 16×16×256 | 37.2 | 11.7 |
| Functa [4] | 32×32×64 | 37.7 | 8.8 |
| LDM [30] | 64×64×3 | 27.4 | 3.6 |

In addition, although we do not conduct exhaustive hyperparamter search, the FID score of generated images achieves 9.3 with classifier-free guidance scale [16] in Table 5. Thus, the results validate the potential applications of the local latents for INRs. Meanwhile, a few generated images may exhibit checkerboard artifacts, particularly in simple backgrounds, but we leave the elaboration of a diffusion process and sampling techniques for generating INR latents as future work.

### 4.5 Comparison with Overfitted INRs

Figure 7 shows that our generalizable INR efficiently provides meaningful INRs compared with individual training of INRs per sample. To evaluate the efficiency of our framework, we select ten images of FFHQ 256×256 and train randomly initialized FFNet [36] per sample using one NVIDIA V100 GPU. The individual training of FFNets requires over 10 seconds of optimization to achieve the same PSNRs of our framework, where our inference time is negligible. Moreover, when we apply the test-time optimization (TTO) only for the extracted latents, it consistently outperforms per-sample FFNets for 30 seconds while maintaining the structure of latents. When we consider the predicted INR as initialization and finetune all parameters of the INR decoder per each sample, our framework consistently outperforms the per-sampling training of INRs from random initialization.

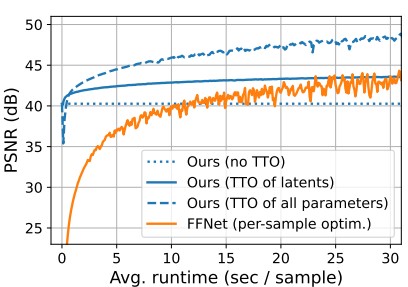

Figure 7: Comparison with individually trained FFNets [36] per sample.

Thus, the results imply that leveraging generalizable INR is computationally efficient to model unseen data as INRs regardless of a TTO.

## 5 Conclusion

We have proposed an effective framework for generalizable INR with the Transformer encoder and locality-aware INR decoder. The Transformer encoder capture the locality of data entities and learn to encode the local information into different latent tokens. Our INR decoder selectively aggregates the locality-aware latent tokens to extract a modulation vector for a coordinate input and exploits the multiple bandwidths of frequency features to effectively predict the fine-grained data details. Experimental results demonstrate that our framework significantly outperforms previous generalizable

INRs on image reconstruction and few-shot novel view synthesis. In addition, we have conducted the in-depth analysis to validate the effectiveness of our framework and shown that our locality-aware latent tokens for INRs can be utilized for downstream tasks such as image generation to provide realistic images. Considering that our framework can learn the locality in non-grid coordinates, such as the Plücker coordinate for rays, leveraging our generalizable INR to generate 3D objects or scenes is a worth exploration. In addition, extending our framework to support arbitrary resolution will be an interesting future work. Furthermore, since our framework has still room for performance improvement of high-resolution image reconstruction, such as $1024 \times 1024$, we expect that elaborating on the architecture and techniques for diffusion models to effectively generate INRs is an interesting future work.

## 6 Acknowledgements

This work was supported by Institute of Information & communications Technology Planning & Evaluation(IITP) grant funded by the Korea government(MSIT) (No.2018-0-01398: Development of a Conversational, Self-tuning DBMS, 35%; No.2022-0-00113: Sustainable Collaborative Multimodal Lifelong Learning, 30%) and the National Research Foundation of Korea(NRF) grant funded by the Korea government(MSIT) (No. NRF-2021R1A2B5B03001551, 35%)

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

# A    Implementation Details

We describe the implementation details of our locality-aware generalizable INR with the Transformer encoder and locality-aware INR decoder. We implement our framework based on the official open-sourced implementation of IPC[3] for a fair comparison. Our Transformer encoder comprises six blocks of self-attentions with 12 attention heads, where each head uses 64 dimensions of hidden features, and $R = 256$ latent tokens for all experiments. We use the Adam [20] optimizer with $(\beta_1, \beta_2) = (0.9, 0.999)$ and constant learning rate of 0.0001. The batch size is 16 and 32 for image reconstruction and novel view synthesis, respectively.

## A.1    Image Reconstruction

**178×178 image reconstruction**    For the image reconstruction of CelebA, FFHQ, and ImageNette with 178×178 resolution, we use $L = 2$ level of modulation features for multi-band feature modulation of locality-aware INR decoder. The dimensionality of frequency features and hidden layers in the INR decoder is 256, where $(\sigma_1, \sigma_2, \sigma_q) = (128, 32, 16)$. We represent a 178×178 resolution of the image as 400 tokens, where each token corresponds to a 9×9 size of the image patch with zero padding. We use a multi-head attention block with two attention heads for our selective token selection via cross-attention. Following the experimental setting of previous studies [8, 19], we train our framework on CelebA, FFHQ, and ImageNette during 300, 1000, and 4000 epochs, respectively. When we use four NVIDIA V100 GPUs, the training takes 5.5, 6.7, and 4.3 days, respectively.

**ImageNet 256×256**    We use $L = 2$ level of feature modulation for the image reconstruction of ImageNet with 256×256 resolution. We use eight heads of selective token aggregation, 256 dimensionality of frequency features and hidden layers of the INR decoder, and $(\sigma_1, \sigma_2, \sigma_q) = (128, 32, 16)$. An image is represented as 256 tokens, where each token corresponds to a 16×16 patch in the image. We use eight NVIDIA A100 GPUs to train our framework on ImageNet during 20 epochs, where the training takes about 2.5 days.

**FFHQ 256×256, 512×512, and 1024×1024**    Our framework for FFHQ 256×256 and 512×512 uses $L = 2$ level of feature modulation with $(\sigma_1, \sigma_2, \sigma_q) = (128, 32, 16)$. The size of each patch is 16 and 32 for 256×256 and 512×512 resolutions, respectively, the number of latent tokens is $R = 256$, and the dimensionality of the INR decoder is $d_F = d = 256$. Our selective token aggregation uses two and four heads of cross-attention for FFHQ 256×256 and 512×512, respectively. We randomly sample the 10% of coordinates to be decoded at each training step to increase the efficiency of training. We train our framework during 400 epochs, while the training takes about 1.5 days using four NVIDIA V100 GPUs for FFHQ with 256×256 and about 1.4 days using eight V100 GPUs for FFHQ with 512×512. For FFHQ 1024×1024, we use 48 patch size to represent an image as 484 data tokens and $L = 2$ level of feature modulation with $(\sigma_1, \sigma_2, \sigma_q) = (256, 64, 32)$. The training of 400 epochs takes about 3.4 days using eight NVIDIA V100 GPUs.

## A.2    Novel View Synthesis

We train our framework for the task of novel view synthesis on ShapeNet Cars, Chairs, and Lamps. Given a few known camera views as support views of a 3D object, our framework predicts a light field of the 3D object to predict unseen camera views. For a fair comparison, we use the same splits of train-valid samples with previous studies of generalizable INR [8, 19, 35]. Given rendering images of support views with 128×128 resolution, we first patchify each rendered image into 256 tokens with 8×8 size of patches. Then, we concatenate the patches of all support views with learnable tokens for the input of our Transformer. We use the Plücker coordinate to represent a ray for a pixel as an embedding with six dimensions and concatenate the ray embedding into each pixel along the channel dimension. Since our INR decoder estimates a light field of a 3D object, the INR decoder has six input channels $d_{in} = 6$ for a ray coordinate and three output channels $d_{out} = 3$ for a RGB pixel. Our INR decoder uses $L = 2$ level of feature modulation with $(\sigma_1, \sigma_2, \sigma_q) = (8, 4, 2)$. We use $d_F = d = 256$ dimensionality of the frequency features and hidden features of the INR decoder. We use 1000 training epochs for ShapeNet Cars and Chairs, while using 500 epochs for ShapeNet Lamps.

---

[3]https://github.com/kakaobrain/ginr-ipc

### A.3 Diffusion Model for INR generation

We implement a diffusion model to generate the latent tokens for INRs of ImageNet 256×256. Different from the conventional approaches, which use a U-Net architecture to generate an image, we use a vanilla Transformer with a simple stack of self-attentions, since the latent tokens do not predefine 2D grid structure but are permutation-equivariant. The Transformer for the diffusion model has 458M parameters having 24 self-attention blocks with 1024 dimensions of embeddings and 16 heads. We remark that the locality-aware generalizable INR is not updated during the training of diffusion models. For the training of the diffusion model, we follow the formulation of DDPM [15]. The linear schedule with $T = 1000$ is used to randomly corrupt the latent tokens for INRs using isotropic Gaussian noises, and then we train our Transformer to denoise the latent tokens. Instead of the $\epsilon$-parameterization that predicts the noises used for the corruption, our Transformer $\mathbf{x}_0$-parameterization to predict the original latent tokens. We drop 10% of class conditions for our model to support classifier-free guidance following the conventional setting [16]. For the stability of training, we standardize the features of latent tokens, after computing the mean and standard deviation of feature channels of each latent token based on the training data. We use eight NVIDIA A100 GPUs to train the model with 256 batch size during 400 epochs, where the training takes about 7 days. The Adam [20] optimizer with constant learning rate 0.0001 and $(\beta_1, \beta_2) = (0.9, 0.999)$ is used without learning rate warm-up and any weight decaying. During training, we further compute the exponential moving average (EMA) of model parameters with a decaying rate of 0.9999. During the evaluation, we use the EMA model with 250 DDIM steps [33] and 2.5 scales of classifier-free guidance [16].

## B   Additional Experiments

### B.1   Ablation Study on the Number of Levels

Table 6 demonstrates the effect of the number of levels $L$ on image reconstruction benchmarks of FFHQ images with 256×256, 512×512, and 1024×1024 resolutions. Our INR decoder uses bandwidths $\sigma_{\mathrm{q}} = 16$ and $(\sigma_\ell)_{\ell=1}^L$ equal to $(128)$, $(128, 32)$, $(128, 64, 32)$ and $(128, 90, 64, 32)$ for $L = 1, 2, 3, 4$ respectively in case of 256×256 and 512×512 resolution, and all bandwidths are doubled for 1024×1024 to leverage high-frequency details.

Table 6: PSNRs on the reconstructed FFHQ with 256×256, 512×512, and 1024×1024 resolutions for different number of levels.

|  | 256×256 | 512×512 | 1024×1024 |
|---|---|---|---|
| TransINR | 30.96 | 29.35 | - |
| IPC [19] | 34.68 | 31.58 | 28.68 |
| Ours ($L = 1$) | 37.09 | 34.84 | 31.56 |
| Ours ($L = 2$) | 39.88 | 35.43 | 31.94 |
| Ours ($L = 3$) | **40.13** | **35.58** | **32.40** |
| Ours ($L = 4$) | 39.79 | 35.40 | 32.32 |

Note that our framework outperforms previous studies [8, 19] even with $L = 1$. Moreover, the results demonstrate that increasing $L$ improves the performance, while the performance saturates beyond $L \geq 3$. We postulate that higher resolution requires a larger number of levels, as the performance gap between $L = 3$ and $L = 4$ decreases as the resolution increases.

### B.2   Additional Examples of Novel View Synthesis

In Figure 8, we show additional examples of novel view synthesis of ShapeNet Chairs, Cars, and Lamps with one to five support views.

### B.3   Additional Examples of High-resolution Image Reconstruction

Figure 9 and 10 shows image reconstruction examples of FFHQ with 256×256, 512×512, and 1024×1024 resolution by previous studies [8, 19] and our locality-aware generalizable INR. Unlike previous studies, our framework can successfully reconstruct fine-grained details in high resolutions.

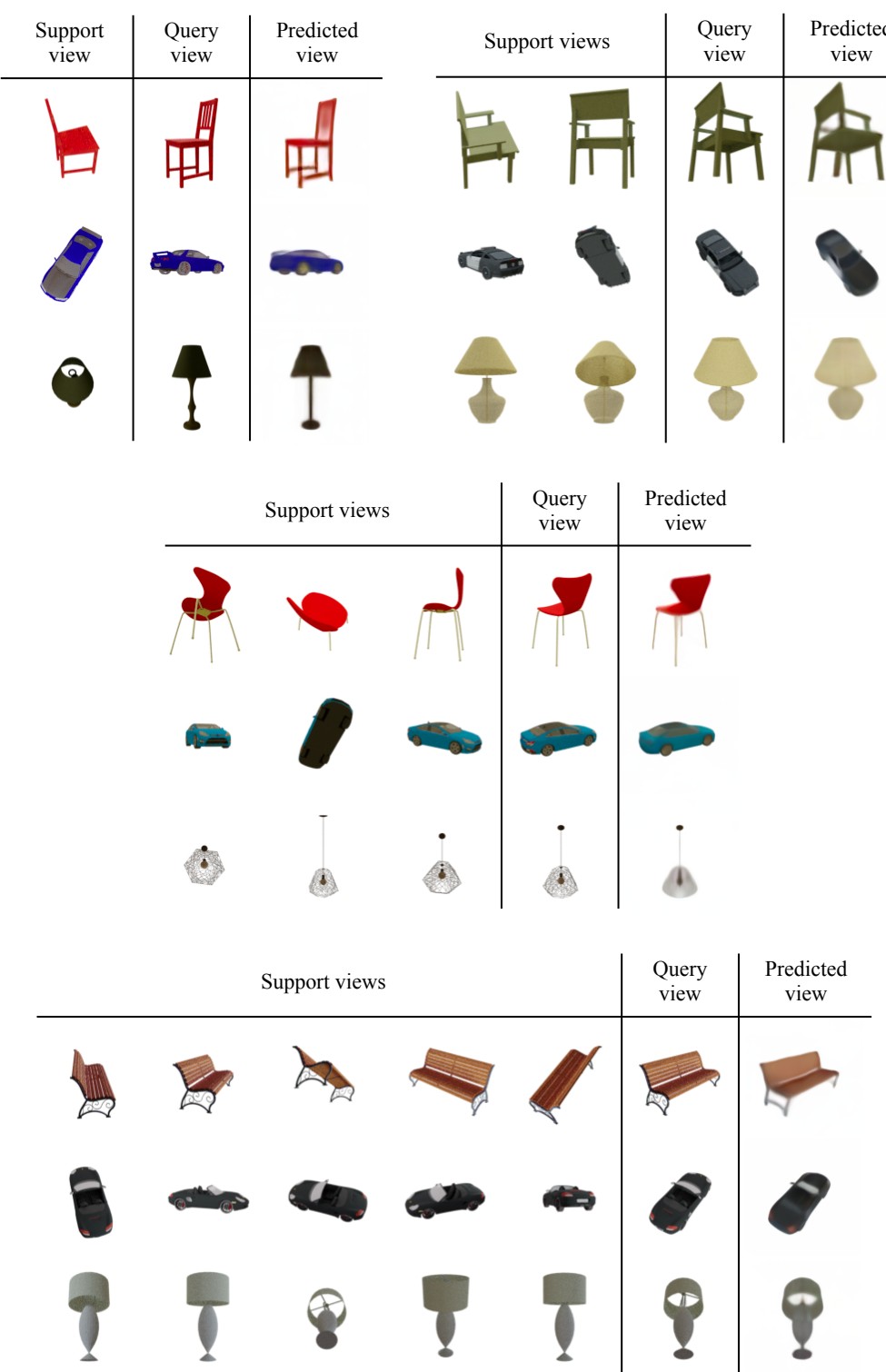

Figure 8: Examples of novel view synthesis of ShapeNet Chairs, Cars and Lamps with one, two, three, and five support views.

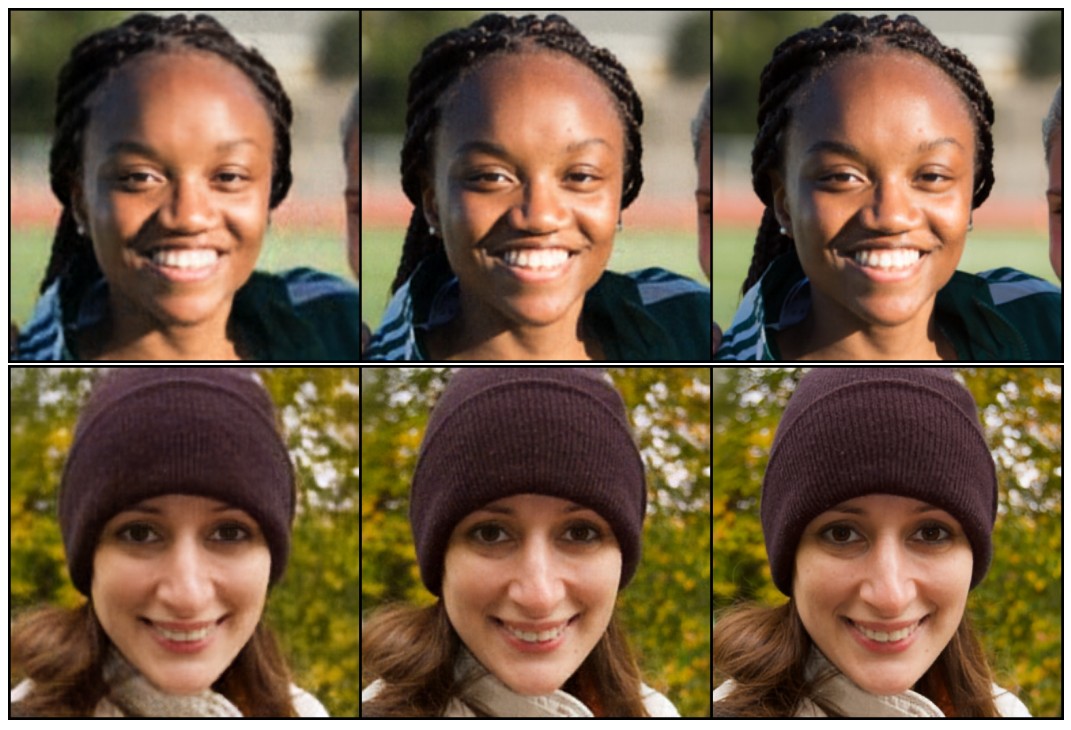

Figure 9: Examples of reconstructed images of FFHQ with 256×256 resolution (top row) and 512×512 resolution (bottom row) by TransINR [8] (left), IPC [19] (middle), and our locality-aware generalizable INR (right).

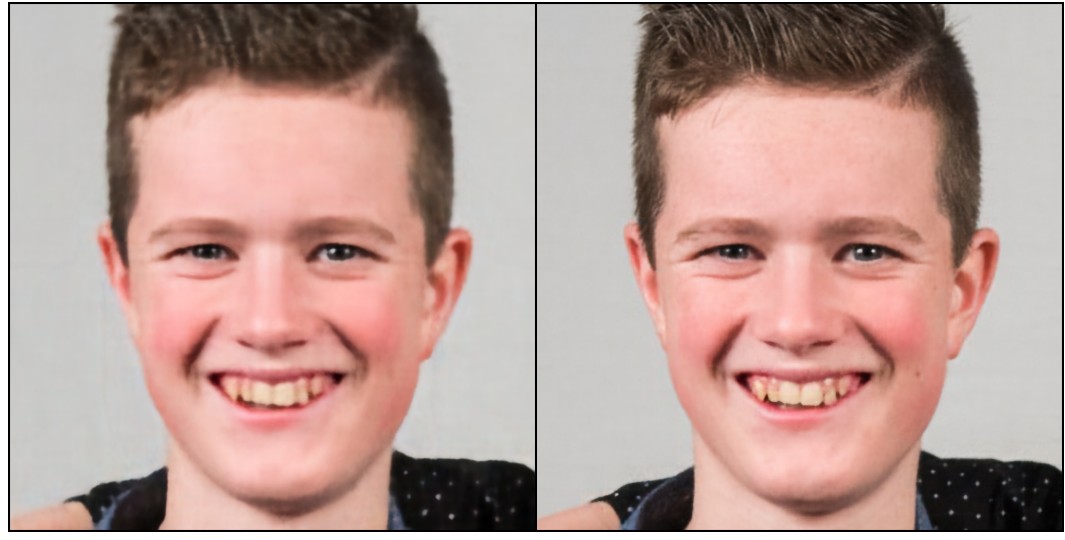

Figure 10: Examples of reconstructed images of FFHQ with 1024×1024 resolution by IPC (left) and our locality-aware generalizable INR (right).

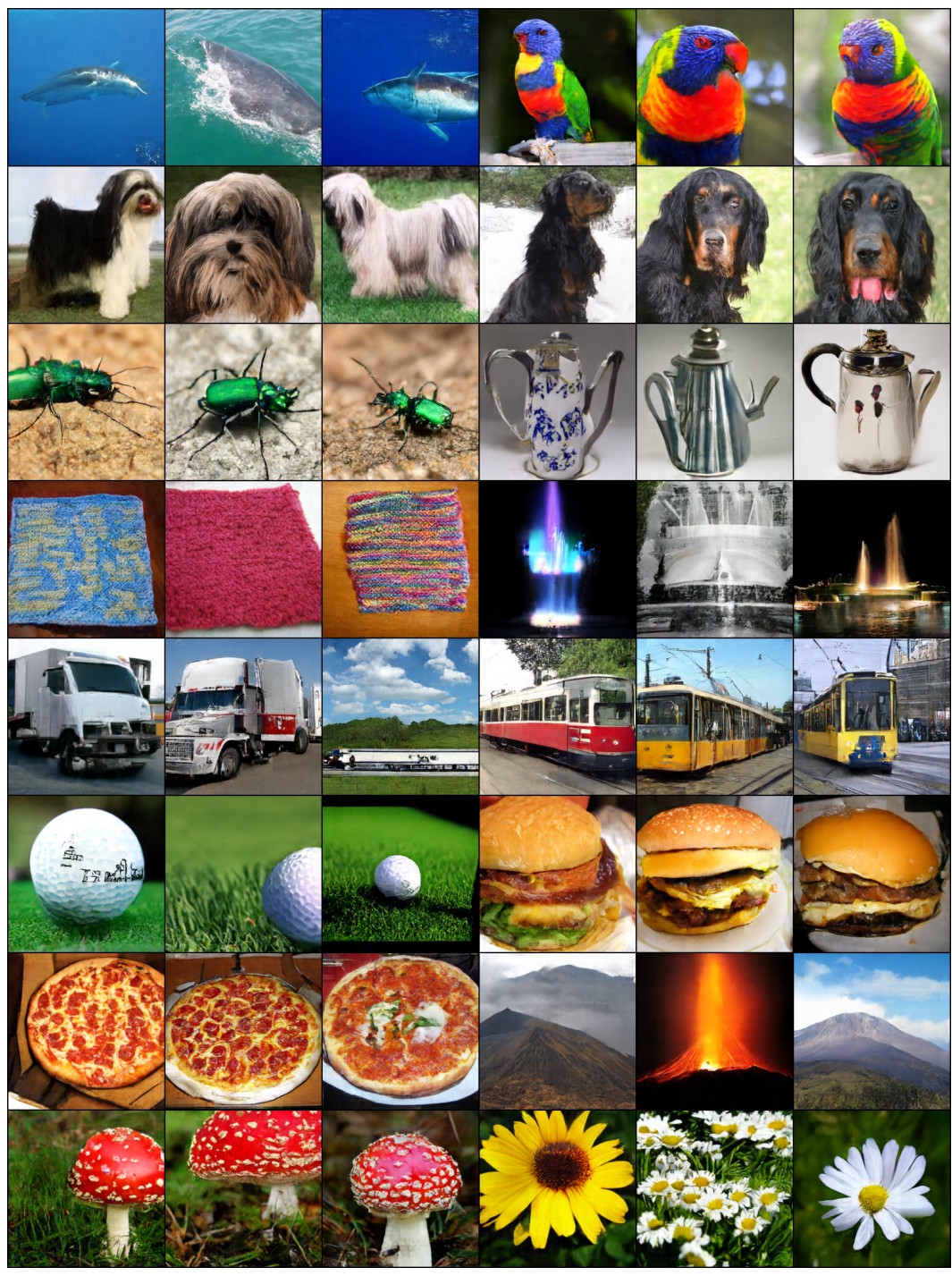

Figure 11: Additional examples of class-conditional image synthesis by generating the locality-aware latents of our framework via a transformer-based diffusion model with 458M parameters. All images are generated with classifier-free guidance at scale 2.5.

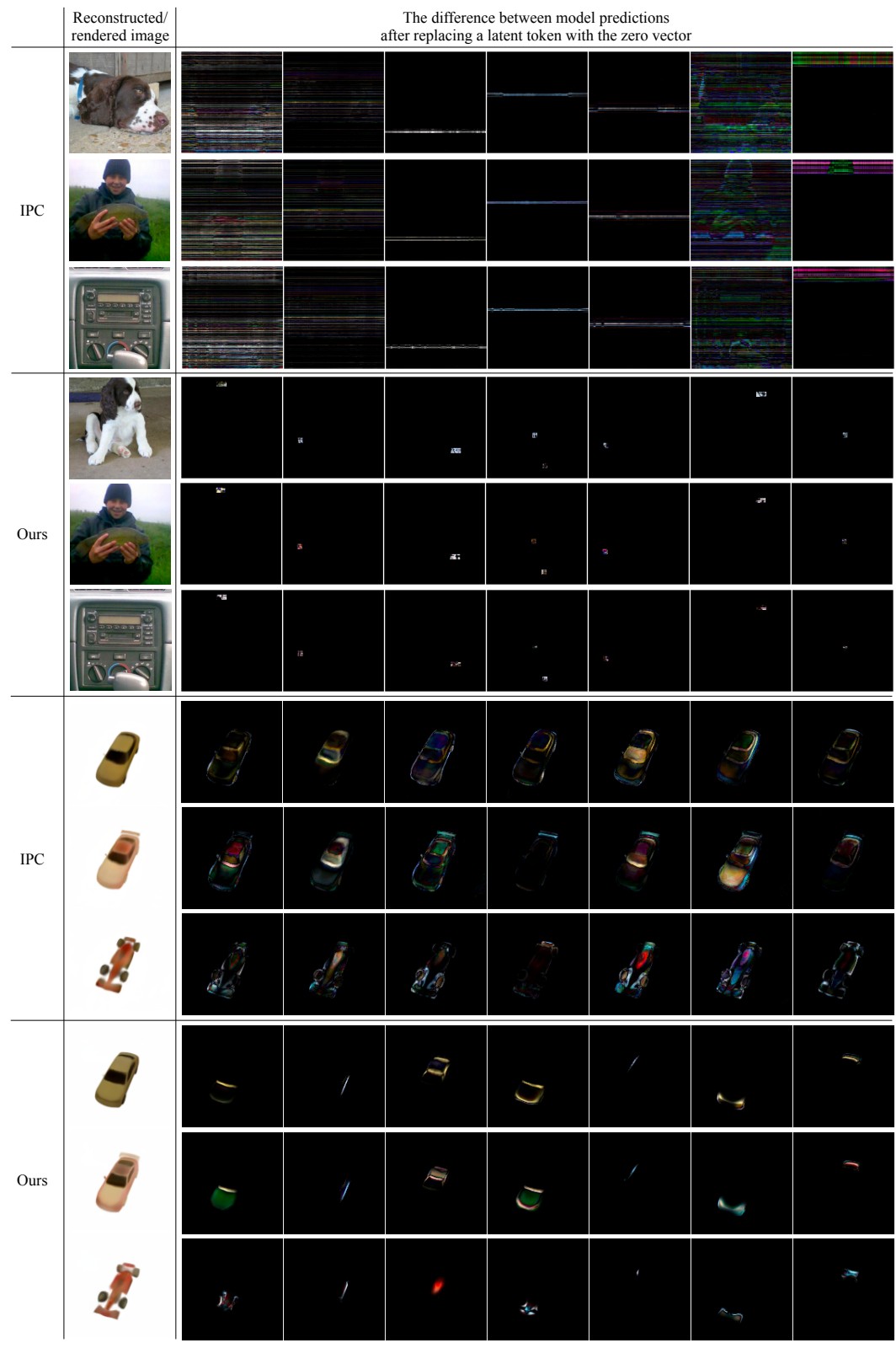

Figure 12: Additional visualization of differences between model predictions after replacing a latent token with the zero vector for IPC [19] and our framework.

### B.4 Additional Examples of Conditional Image Synthesis

Figure 11 shows additional examples of generated images with 256×256 resolution by generating locality-aware latents of our framework.

### B.5 Additional Visualization for Locality Analysis

Figure 12 visualizes which local information of data is encoded in each latent token of IPC [19] and our locality-aware generalizable INR in addition to Figure 5. We randomly select a latent token and replace it with the zero vector, then visualize the difference between the model predictions with or without the replacement as described in Section 4.3. The differences are rescaled to have the maximum value of 1 for clear visualization. Furthermore, we fix the set of replaced latent tokens for different samples in Figure 12 to emphasize the role of each latent token. Note that each latent token of our framework encodes the local information in a particular region of images or light fields, while latent tokens of IPC encode global information over whole coordinates.

### B.6 Ablation Study on Linear Layers in Selective Token Aggregation

Our framework adds a linear layer in Eq (6) and Eq (7) to exploit complex frequency patterns, improving the performance. While the Fourier features consist of periodic patterns along an axis, the frequency patterns in Eq (6) can also include non-periodic patterns. Note that IPC [19] also uses a similar design, while modulating the second MLP layer to exploit complex frequency patterns. The linear layer in Eq (7) is used to process the modulation vector according to each frequency bandwidth, motivated by the design of separate projections for (query, key, value) in self-attention. The results below also show that removing the linear layers in Eq (6) and Eq (7) significantly deteriorates the image reconstruction performance on ImageNette 178×178.

Table 7: PSNRs on the reconstructed ImageNette with 178×178 resolution.

|  | PSNR |
|---|---|
| Ours | **37.46** |
| w/o Linear in Eq (6) | 31.95 |
| w/o Linear in Eq (7) | 32.07 |
| w/o Linear in Eq (6) and Eq (7) | 31.57 |

