# OpenReview forum: "Locality-Aware Generalizable Implicit Neural Representation"
_NeurIPS.cc/2023/Conference — NeurIPS 2023 poster_

### Official Review · Reviewer_9zhe · 2023-07-04

**Soundness:** 3 good
**Presentation:** 3 good
**Contribution:** 3 good
**Rating:** 6
**Confidence:** 3

**Summary:**

This paper studies the problem of generalizable implicit neural representation. The proposed method combines a transformer encoder with a locality-aware decoder to predict the output with the feature modulation through multiple frequency bandwidths. Experiments are performed on image reconstruction, few-shot novel view synthesis, and image synthesis to show the effectiveness of the proposed method for generalizable implicit neural representation.

**Strengths:**


- This paper studies an important problem in implicit neural representation and the motivation is clear.

- This paper is generally well structured and the proposed method is easy to follow.

- Experiments are sufficient to show the effectiveness of the proposed method.


**Weaknesses:**


- This paper mainly emphasis on locality-aware, but fails to provide a formal definition of the notation "locality". It would be better to provide a formal formulation and some intuitive examples.

- Many recent works tend to explore a hybrid neural representations (e.g., feature maps with small mlps), while this paper only considers the pure MLPs. It would be interesting to have a further discussion about this.

**Questions:**

See weakness.

**Limitations:**

The authors have adequately addressed the limitations.

---

> ### Author Rebuttal · Authors · 2023-08-09
>
> **[Locality: Definition and Intuitive Examples]**
> We will include the following additional explanation of the concept of locality for a better understanding of readers. Although it is challenging to provide a formal definition of ‘locality’, we can provide its conceptual description and intuitive examples for clear understanding. Note that the term locality in the operating system field is also informally described. In our study, ’locality’ is a concept that the features/attributes of data points are highly correlated when these points in the input space are located nearby, having a small distance between data points. For example, two nearby pixels in a 2D image have potentially similar colors. Thus, our framework learns to extract a latent corresponding to specific local regions for effective data representation, leveraging the inherent high correlation of features within local regions.
>
>
> **[Discussion about a Hybrid Neural Representation]**
> Our framework and recent approaches of a hybrid neural representation commonly extract latents from local regions to predict the features on continuous coordinates. However, our framework does not assume the explicit grid structure of local latents, unlike the recent hybrid neural representations that assume an explicit grid structure of data (e.g., 2D grid for an image or 3D grid for a 3D object) to extract the latents of each region [NewRef: InstantNGP]. Thus, our framework can be viewed as a broader version of hybrid neural representations, capable of learning the local structure across diverse data types.
>
> [NewRef: InstantNGP] Müller, Thomas, et al. "Instant neural graphics primitives with a multiresolution hash encoding." ACM Transactions on Graphics (ToG) 41.4 (2022): 1-15.

---

> > ### Comment · Reviewer_9zhe · 2023-08-18
> >
> > Thank you for your response and it helped clear up some of my concerns. I would like to keep my original rating.

---

### Official Review · Reviewer_binY · 2023-07-05

**Soundness:** 3 good
**Presentation:** 4 excellent
**Contribution:** 4 excellent
**Rating:** 7
**Confidence:** 3

**Summary:**

Generalizable implicit neural representation (INR) can represent multiple data instances with a coordinate-based neural network, of which weights or intermediate features are modulated using instance-wise latent codes. A significant constraint of current generalizable INR is their struggle to localize and capture fine-grained details of data entities. This limitation results in a diminished expressive power of modulation.

This research tackles this problem by presenting an innovative framework for a generalizable Implicit Neural Representation (INR). This framework merges a transformer encoder with a locality-aware INR decoder. Two key components are designed to enable to capture of the local information of data, including selective token aggregation and multi-band feature modulation.

The authors demonstrate the effectiveness of their method with state-of-the-art performances on both image reconstruction and novel view synthesis. Furthermore, they show the potential of the proposed generalizable INR on the conditional image synthesis task.


**Strengths:**

- **Scope and relevance**: This paper studies an important topic that enables INR with generalization ability.

- **Technical contribution**: The authors present a novel locality-aware INR decoder to improve the expressive power of modulation by learning locality-aware representation from data.

- **Experiments**: The authors conduct extensive experiments and demonstrate significantly improved performances on multiple benchmarks.

- **Clarity**:  The paper is well-written.

**Weaknesses:**

Some technical details might need further examination.

- **Selective Token Aggregation**: In Section 3.3.1, the cross-attention is applied in extracting a modulation vector $\mathbf{m}_\mathbf{v}$  with the coordinate $\mathbf{v}$ and the latent tokens. Can the cross-attention be replaced with standard transformer attention with input from the concatenation of latent tokens and the frequency features of the coordinate? Can other operations that fusion these two inputs also work well?

- **Multi-Band Feature Modulation**: The authors use a range of frequency bandwidths to predict the details of outputs with a deeper MLP path to learning higher-frequency features. If I am not mistaken, more learnable parameters are introduced into the coordinate-based neural network. Is it possible the improved performance stem from the increased model size?


**Questions:**

In addition to the above weaknesses, here are two more questions about conditional image synthesis with the extracted latent tokens (Section 4.4 and Appendix A.3):

- To realize conditional image synthesis, do the authors mean by training a diffusion model to generate latent tokens with pre-trained generalizable INR models?

- In lines 522-523, "We drop 10% of class conditions for our model to support classifier-free guidance." Why 10%? Does this number percentage matter?

Overall, this paper is a good effect. I will raise my rating if the authors can address my concerns.

**Limitations:**

The authors haven't adequately addressed limitations and the potential negative societal impact of their work.

I would suggest showing some failure examples, in which the proposed model can not work well, so as to let the community know the boundary of this work.

Moreover, considering the critical importance of AI safety, it could be beneficial to discuss any potential adverse societal impacts that may arise from this work.

---

> ### Author Rebuttal · Authors · 2023-08-09
>
> **[Standard Transformer Attention for Selective Token Aggregation]**
> Although the standard transformer’s self-attention can be used to predict the modulation vector for a coordinate input, we adopt cross-attention due to the computational efficiency. Using self-attention with the concatenation of latent tokens with the frequency features of the coordinate results in the computational costs with $O( (R+1)^2)$, where $R$ denotes the number of localized latents. Contrastively, using cross attention requires $O(R \times 1)$. While other operations, such as k-NN, can also be used for selective token aggregation, using cross-attention is a natural and intuitive design choice within a modern deep learning architecture.
>
>
> **[Multi-Band Feature Modulation]**
> In Section 4.1, the number of trainable parameters for our framework is 44.14M, while IPC has 43.75M parameters. That is, the increased number of parameters is 0.4M, which is 0.9% parameters of the IPC framework. Although our framework uses 0.9% more total parameters than IPC, the improved performance is significant and not dependent on the increased parameters. To be precise, we increase the hidden dimension of MLP, $d$, for IPC and train the framework of IPC on FFHQ 256x256, as shown in the table below. Increasing the number of MLP parameters also improves the performance of IPC. However, the results show that our performance improvement does not come from a simple increase of trainable parameters but from effective architecture design.
>
> | FFHQ 256x256  | # parameters | d |  R | PSNR |
> |:------|:------:|:------:|:------:|:------:|
> |Ours | 44.14 M | 256 | 256 | **39.88** |
> |IPC [19] | 43.75 M | 256 | 256 | 34.68 |
> |IPC [19] | **65.31 M** | **1024** | 256 | 38.43 |
>
>
>
> **[Training a Diffusion Model]**
> Yes, we utilize a pre-trained generalizable INR model to train a diffusion model for image generation. Specifically, after our generalizable INR is trained on ImageNet 256x256 to extract the localized latents of the images, the extracted latents of images are randomly corrupted by Gaussian noises to train a diffusion model.
>
>
> **[10% Drop of Class Conditions]**
> We follow the original paper on classifier-free guidance and conventional settings to train class-conditional diffusion models. While we have not conducted an ablation study on the probability of unconditional training (class condition drop), the original paper on classifier-free guidance describes the following findings: when $p_\text{uncond} \in \{ 0.1, 0.2, 0.5 \}$ is ablated, $p_\text{uncond}=0.5$ consistently performs worse than $p_\text{uncond} \in \{ 0.1, 0.2 \}$. Thus, it has been concluded that only a small portion of the model capacity is needed for the unconditional generation task for classifier-free guidance.
>
> **[Limitations]**
> We will add the following discussion about the limitations of our study. Although our framework significantly improves the image reconstruction performance for INRs, there is still room for improvement, especially in high-resolution image reconstruction, such as 1024x1024. Our experiments on novel view synthesis have been conducted only on a category-specific and synthetic dataset such as ShapeNet. Since our framework can be applied to generation tasks, our study is not exempt from generative models' potential negative societal effects.

---

> > ### Comment · Reviewer_binY · 2023-08-11
> > **Rebuttal Acknowledgment**
> >
> > I thank the reviewers for their detailed clarification, which has released my concerns about their work. Therefore, I raise my rating to accept.

---

> > > ### Author Response · Authors · 2023-08-11
> > > **Thank you for engaging in the discussion and increasing the score.**
> > >
> > > Dear reviewer,
> > >
> > > Thank you for engaging in the discussion and increasing the score.
> > > We sincerely appreciate your time and dedication in this matter.

---

### Official Review · Reviewer_KNfA · 2023-07-06

**Soundness:** 1 poor
**Presentation:** 2 fair
**Contribution:** 2 fair
**Rating:** 3
**Confidence:** 5

**Summary:**

This paper enhances the performance of generalizable INR via improving its locality awareness. The transformer encoder feed on patchs of an image and produce latent tokens with local information, and later extracted as modulation vectors. The proposed INR decoder with selective token aggregation and the multi-band feature modulation learn locality-aware representation in spatial and spectral domains respectively. Experiments show that the proposed method outperform previous methods both on reconstruction and downstream image generative tasks.

**Strengths:**

1. The proposed make a lot of sense to enhance the locality of INRs and via patch processing and attentions in transformers etc., and it reaches very good results compared to previous SOTAs. The method is clearly illustrated, and the results are well displayed both quantitatively and qualitatively. There are also many ablations and discussions/analyses into the details and depth. The supplementary material also provides more details, discussions and results.

2. Besides the main reconstruction task, the proposed method is also able to perform a few generative tasks well, and also better than previous SOTAs.

3. Figure 5 (and Figure 12 in Supp) shows interesting results on replacing latent tokens with null ones, unvealing some underlying rationales of the learned space.

4. Source code is provided in the supplementary material.

**Weaknesses:**

1. [Core] An INR usually refers to a neural network that store information in its model weights instead of feature maps. The proposed modulation vector m_v seems to be more like a latent code (in an encoder-decoder framework). I hope to see clear clarification on how the parameters of the proposed INR is composed of, i.e. how much portion is feature maps and how much portion is model weights. In addition, if most (or almost all) parameters of the "INR" is just feature maps, then it might not be very difficult to perform those downstream generative tasks, as it is essentially more an improved VAE than an independent INR. I'm wondering how this could help the displayed performance.

2. [Core] The INR (decoder or modulation vectors, i.e. the non-shared parts to independently form a single INR representing a single image) model size / compression ratio seems not to be mentioned in the paper. How big the produced INR is to representing one image of 256x256 for example? The paper has conducted a few ablations on the model structure details, including Table 4 and Table 6, while I'm wondering a more overall metric, such as number of paramters in an INR, its size in MB, or BPP (bits per pixel). How does this compared to other methods such as TransINR ordan IPC?

(The above two points are my main concerns and the game-changer for my assessment.)

3. Figure 2 shows the core mechanism clearly, but in addition I think a specific figure to show the produced INR itself (e.g. many layers or blocks duplicated) would make the full pipeline clearer to understand.

4. Table 5 seems not to be mentioned in the main body text nor illustrated anyhow (while L271 should be referring to Figure 4 as a typo).

**Questions:**

1. The provided ablations are on reconstruction performance only. While this is the main task and objective, I'm also wondering how these ablations perform on other downstream applications in the paper, including novel view synthesis and image generation. Does any options or hyperparameters make a difference for these applications? Or do you find you don't have to carefully pay attention to them in these applications in your experiments?

**Limitations:**

1. More discussions and analyses on the learned INR model weight space would be welcomed and inspiring. How do you think the locality awareness may improve the INR space distance smoothness? For example, in LDM they applied KL loss or VQ regularization on VAE (although not big) to help the distance in the latent space (feature map) be consistent with that in the pixel space. Do you think this might be one of the point that the proposed method reaches better results in generative tasks including novel view synthesis and image generation?

2. IPC also conduct experiments on downstream classification tasks. It would be great to see similar results on your proposed method and comparisons with them. (Or do you think there might be any difference in design making that your method is better at and more fitful for generative tasks than classifications? Any discussion is welcomed.)

---

> ### Author Rebuttal · Authors · 2023-08-09
>
> **[The Portion of Localized Latents in an INR]**
> The localized latents account for 9% of INR parameters (=65,536/725,446), which is a small portion of INR. Since the shape of extracted localized latents is 256x256, we emphasize that the size of localized latents is equivalent to **one weight matrix of a single MLP layer** with 256 hidden dimensions.
> Meanwhile, as the reviewer comments, our framework can also be interpreted as an improved version of encoder-decoder (INR) architecture, since we also use a hypernetwork to predict the latents for modulating INRs. A conventional VAE has a decoder of substantial size, having tens of millions of parameters (>10M), which is used to generate or reconstruct an image from feature maps. However, our INR decoder has a much smaller number of parameters, (0.73M) with its role being to simply represent a data instance itself as INR.
>
>
> **[The Size of Non-Shared Part]**
> The size of non-shared latents is fixed to 256x256 for all experiments for TransINR, IPC, and our framework, including experiments on high resolution images such as FFHQ 1024x1024. Thus, TransINR, IPC, and our framework require 0.25MB (=32-bit x 256 x 256) to specify a data instance as an INR. The compression ratio is 130%, 33%, and 8% for 256x256, 512x512, and 1024x1024 images, respectively. Compared with a previous study (e.g., COIN [10]), which exploits an INR for data compression and combines quantization techniques to improve compression ratio, generalizable INRs show worse data compression performance, since they have focused on improving the reconstruction performance. Considering that our framework significantly improves the reconstruction performance of a generalizable INR, combining quantization techniques with localized latents will be an interesting future work for data compression. We will attach this discussion to our revised manuscript.
>
>
> **[Explanation about Figure 2]**
> Figure 2 shows the produced INR in our experiments, illustrating the number of duplicated layers or blocks. We will add a detailed explanation to ensure understanding.
>
> **[Referring Table 5]**
> Thanks for the detailed comment. We will change “Figure 5” in line 271 to “Figure 6” and explicitly mention Table 5 in Section 4.4.
>
>
> **[Ablation Studies on Novel View Synthesis and Image Generation]**
> Since training a diffusion model for image generation is expensive, we also attach the ablation study on the novel view synthesis of ShapeNet-Lamp with 3 support views. The results below show that both selective token aggregation (STA) and multi-band frequency modulation (multiFM) improve the performance of our framework.
>
> | | ImageNette | FFHQ | Lamps-3 views |
> |:------|:------:|:------:|:------:|
> |Ours | 37.46 | 38.01 | 26.00 |
> |w/o STA | 34.54 | 34.52 | 25.35 |
> |w/o multiFM | 33.90 | 33.65 | 25.78 |
> |IPC [19] | 34.11 | 34.68 | 25.09 |
>
>
>
>
> **[More Discussions and Analyses on the Learned INR]**
> As the reviewer’s comment, we believe that the locality awareness can also enhance the smoothness of the INR space. Figure 6 in our submission shows that the generated latents of IPC cannot provide realistic local details, since an artifact affects all coordinates, as described in Lines 303-305 and Figure 5. The results imply that the INR space of IPC is sensitive to slight variations or corruptions, while our framework has a more smooth INR space than IPC.
>
> Meanwhile, the regularization techniques (e.g., KL loss or VQ regularization), which the reviewer has mentioned, can also be applied to our framework, making the INR space further smoother. However, when employing the KL regularization with 1E-6 loss weight, the PSNR on ImageNet 256x256 decreases from 37.7 to 30.7. We have also changed the KL regularization in LDM to the deterministic version and fixed the predicted covariance as Isotropic Gaussian, then the reconstruction PSNR becomes 33.9. We believe that further exploration of hyperparameters can enhance our framework, expecting that a smoother INR space can help a diffusion model improve the performance of downstream tasks.
>
>
>
> **[Experiments on Downstream Classification Tasks]**
> We have checked that IPC does not conduct an experiment on any downstream task, but Spatial Funta [4] has conducted an experiment on a downstream classification task. We have yet been able to conduct the classification task due to the limited computational resources during the rebuttal period. However, we expect similar results with Spatial Functa, since the performance of generative tasks also shows similar results with Spatial Functa. We do not assume that our framework is more fitful for generative tasks than classification. Instead, we perceive a generative task to be more difficult than classification. Note that a generative task requires learning all local data details, while a classification task requires understanding the global semantics. We will do our best to reserve additional computational resources for conducting experiments on classification tasks to attach the results in the final version.

---

> > ### Comment · Reviewer_KNfA · 2023-08-15
> >
> > Thank you for your clear and informative responses.
> >
> > If I'm understanding it correctly, is the localized latent vector the only non-shared part for each INR? Is the decoder shared for all INRs or does each INR has its own decoder, whose weights are generated by the transformers given the specific image?
> >
> > I appreciate the authors' explanations for my Q1 and Q2; however, I still have distance from completely understanding it clearly. For example, In Q1 it is said that the localized latent is only part of the INR (65,536/725,446). But in Q2 it uses 32-bit x 256 x 256 to calculate the whole size of an INR for the compression ratio. In the main paper L103 it is said that "A generalizable INR uses a single coordinate-based MLP as a shared INR decoder $F_\theta$". I'd appreciate the authors if you could provide more detailed illustrations.
> >
> > In Figure 2, does the shallow yellow background indicate the whole components of one INR? In this way it seems that the decoder is dedicated for each INR? But it doesn't show how the parameters of it (e.g. the Fourier-FCs) are produced?

---

> > > ### Author Response · Authors · 2023-08-15
> > >
> > > Dear Reviewer KNfA,
> > >
> > > We appreciate your active participation and valuable comments in the ongoing discussion. Please find below our responses to your inquiries. We will also revise the paper accordingly.
> > >
> > > - The shallow yellow background in Figure 2 indicates the whole components of one INR. Here, we remark that the localized latent vectors are the only parts unique to each data instance, while the remaining parts are shared across data instances. Given that an FC represents a linear layer, the parameter count for FCs, which are shared among data instances, corresponds to a weight matrix of dimensions $d \times d$, with $d$ being equal to 256.
> > >
> > > - Although each data instance requires 725,446 parameters (32-bit x 725,446), we use the compression ratio (32-bit x 256 x 256) based on the localized latent vectors with 256x256 size, since the remaining parts of the decoder (725,446-65,536=659,910) are shared across data instances. That is, while generalizable INR uses a single coordinate-based MLP as a shared INR decoder (i.e., the remaining parts of the decoder), it is specified by the localized latent vectors to represent each data instance. That is, to represent $N$ data instances, the number of required parameters is 659,910 + 65,536 x $N$, while conventional INRs require 725,446 x $N$.

---

> > > > ### Comment · Reviewer_KNfA · 2023-08-16
> > > >
> > > > Thank you for your further clarification.
> > > >
> > > > So I want to further double confirm, if the parameters of the (shared) decoder are got via common backpropogation training? (i.e. not via for example hypternetwork-like stuff to learn to generate the parameters directly etc.?)
> > > >
> > > > And then I'm curious, what the difference would be between the proposed INR with common autoencoder (or its variants)? For a common autoencoder (or any encoder-decoder framework model), there is a (shared) encoder to extract some key information from the input into a feature map / latent vector, and then a (shared) decoder to reconstruct the original input fed on this latent vector. The bottleneck latent vector is the representation of each data instance under this framework. Can the main contribution of this paper be described as improving the locality of a transformer and Fourier based autoencoder?

---

> > > > > ### Author Response · Authors · 2023-08-16
> > > > >
> > > > > Dear Reviewer KNfA,
> > > > >
> > > > > Thank you for the insightful queries. Kindly refer to the response provided below to address your questions. We provide further elaboration on the differentiation between our framework and conventional autoencoders as follows. I hope that our contributions establish a novel framework for future endeavors.
> > > > >
> > > > > - Indeed, the parameters of our shared decoder are updated through backpropagation during training. In the context of generalizable INRs, the shared parts are updated via backpropagation during training, while the non-shared parts are updated after training for each data instance.
> > > > >
> > > > > - Our framework cannot be simply described as an improved autoencoder; it can encompass an autoencoder that reconstructs input data using the complete observations of the input data. That is, our framework for image reconstruction can be viewed as an improved autoencoder, which can predict the features of continuous coordinates, without necessitating a substantial size (> 10M parameters) of the conventional decoder. However, our framework for novel view synthesis does not fit the autoencoder framework, since our framework learns to synthesize previously **unseen** views of a 3D object, based on its partial observations from a few support views.
> > > > >
> > > > > - Instead, the main contribution of our paper lies in significantly surpassing the limited capacity of state-of-the-art generalizable INRs as described in Lines 28-37. Our framework achieves this through the intelligent incorporation of localized latents, which possess the same capacity as a single weight matrix in an MLP. It is noteworthy that our framework significantly enhances the performance of a generalizable INR in tasks involving high-resolution image reconstruction and novel view synthesis. Furthermore, it demonstrates the potential of generalizable INRs for downstream tasks, including generative modeling.

---

> > > > > > ### Comment · Reviewer_KNfA · 2023-08-18
> > > > > >
> > > > > > Thank you for your further discussions. After carefully reviewing your replies and re-reading the paper again, I think that there are several critical problems of this paper, including its terminologies and contributions, and I misunderstood some important parts of it when initially reviewing it.
> > > > > >
> > > > > > I would like to formally argue and defend that the proposed method in this paper cannot be defined as a generalizable INR. An implicit neural representation means it stores information of each data instance on its own dedicated model weights, for example NeRF, on the contrary to storing information in latent features (i.e. explicit representation). This paper only store data-instance-wise information in the latent features, as the encoder and decoder are shared across all data instances. A generalizable INR means although an INR is usually acquired by backpropogation for each single data instance, it leverages hypernetwork to generate the model weights of INRs in a feed-forward way directly, so that speed up and generalize dedicated INRs. In this paper there is no such process at all. This is a fundamental problem and is far beyond any ambious zone. This problem can be critically misleading and even if there are any good results, it should't be considered as an INR work for followers.
> > > > > >
> > > > > > The calculation of the model size is another example. The INR sizes for TransINR and IPC are definitely not only 256x256, as their decoders have multiple MLP layers and they're dedicated and different for every data instance. These non-shared decoder layers are thus part of the INR, when calculating the size of one INR, and the compression ratio. And therefore they have hypernetworks to generate the weights of these dedicated MLP layers, unlike this paper whose transformers are only used to produce the latent vectors.
> > > > > >
> > > > > > For most tasks in this paper, they are able to be accomplished with autoencoder frameworks. For example latent diffusion models do what exactly shown in Sec. 4.4. There could be performance difference, but after all this paper is not any INR or generalized INR at all. Not to mention that comparing an autoencoder method with other INR methods is not fair at all and gives misleading assessments.
> > > > > >
> > > > > > For the novel view synthesis task, although there hasn't been any published autoencoder method working on it, this doesn't necessarily prove that this paper isn't an autoencoder framework. This paper does improve this task's performance, but still in an upgraded autoencoder way, instead of an INR way. Emphasizing "unseen" doesn't indicate anything, as vanilla autoencoders are able to generate unseen new images when trained on an image dataset with its well-learned latent space at the bottleneck. Besides, most novel view synthesis tasks are on entire scenes which are more complicated than single synthetic 3D objects, which are more complicated and thus simple autoencoder frameworks are hard to handle. Only solving novel view synthesis task on single synthetic 3D objects doesn't indicate a big capability boost either.
> > > > > >
> > > > > > I have to lower my ratings to clear rejection. And I would like other reviewers and chairs also notice this.

---

> > > > > > > ### Comment · Reviewer_AhKU · 2023-08-18
> > > > > > >
> > > > > > > I agree with much of this, and I think in retrospect it explains some of my confusion with the paper's descriptions of what it's claiming to do. My original review was for weak rejection and I will keep that rating, in the hopes that the authors will revise and resubmit a clearer paper whose descriptions and comparisons to baselines are more appropriate for its method/results.

---

> > > > > > > ### Author Response · Authors · 2023-08-18
> > > > > > >
> > > > > > > We appreciate the reviewer's in-depth discussions, but there are some important misunderstandings about our work. We address them below.
> > > > > > >
> > > > > > >
> > > > > > > **[The Definition of INRs]**
> > > > > > > The reviewer stated that “an implicit neural representation means it stores information of each data instance on its own dedicated model weights, for example NeRF, on the contrary to storing information in latent features (i.e. explicit representation).” We agree with this indeed. Recent work for INR, e.g., SIREN (NeurIPS’20) and Grattarola & Vandergheynst (NeurIPS’22), also defines INRs as neural networks that parametrize implicitly defined functions such that the networks properly map some input, e.g., coordinates $\mathbf{v} \in \mathbb{R}^m$, to some quantity of interest, e.g., RGB value. Our *decoder with localized latents* exactly accords with the definition of INR since it can be viewed as a parameterized neural network that maps an coordinate input to a quantity of interest. Note that localized latents $\mathbf{Z}$ are a part of the INR parameters predicted by a hypernetwork in this view. This also corresponds to the concept of generalizable INR the review mentioned: “A generalizable INR … leverages hypernetwork to generate the model weights of INRs in a feed-forward way directly, so that speed up and generalize dedicated INRs.” In our method, the generated weights (parameters), i.e., local latents, are used to modulate a shared INR decoder to represent a specific data instance as $\Phi(\mathbf{v} | \mathbf{Z})$, making the whole INR network better speed up and generalize to different instances.
> > > > > > >
> > > > > > > In a nutshell, our INR network consists of two parts: (1) instance-specific parameters, i.e., local latents, which are generated by a hypernetwork with an image instance, and (2) instance-agnostic parameters, i.e., shared decoder, which are learned and shared across instances.
> > > > > > >
> > > > > > > Unfortunately, the review appears to consider the local latents as explicit features rather than generated parameters, a part of INR. We think this is a misunderstanding because of the following two reasons.
> > > > > > >
> > > > > > > 1. The scope of neural network parameters often extends to token inputs to attention layers in transformer-based architectures, e.g., a dictionary of learnable embeddings or tokens [NewRef 1-3]. Instead of direct optimization of learnable latent embeddings for each data instance, our hypernetwork, i.e., Transformer encoder, learns to directly predict a set of latents (token parameters) to specify an INR decoder for each data instance. We also note that several recent methods for generalizable INR also exploit a latent embedding in a similar manner to ours. For example, DeepSDF (CVPR’19), Functa (ICML’22) and Rebain et al. (TMLR’22) do not predict the weights of MLP layers, but leverage a latent embedding as an instance-specific parameter to modulate the intermediate features of a shared INR decoder.
> > > > > > >
> > > > > > >
> > > > > > >
> > > > > > >
> > > > > > > 2. Our localized latents are not explicit representation as the latent features of auto-encoders. As the reviewer mentioned, latent features in auto-encoders are explicit representations indeed since they can directly be decoded to a data instance. However, our localized latents require additional data coordinates, which are actual input to the function the localized latents parameterize in part, i.e., INR decoder network. In this sense, the local latents are implicit representation, and they are a part of a decoding function.
> > > > > > >
> > > > > > > We hope these clarifications address the reviewer’s concern. At the same time, we admit that our current manuscript is not very clear on these points, which may have confused the reviewer. We will clarify them with more careful terms and revise our figures, e.g., Fig. 2, to better see the points: hypernetwork, INR structures, and their inputs and outputs.
> > > > > > >
> > > > > > >
> > > > > > > [NewRef 1] Touvron, Hugo, et al. "Training data-efficient image transformers & distillation through attention." International conference on machine learning. PMLR, 2021.
> > > > > > > [NewRef 2] Sandler, Mark, et al. "Fine-tuning image transformers using learnable memory." Proceedings of the IEEE/CVF Conference on Computer Vision and Pattern Recognition. 2022.
> > > > > > > [NewRef 3] Radford, Alec, et al. "Learning transferable visual models from natural language supervision." International conference on machine learning. PMLR, 2021.

---

> > > > > > > ### Author Response · Authors · 2023-08-18
> > > > > > >
> > > > > > > **[Comparison of Model Size]**
> > > > > > > First, in contrast to the reviewer’s comment that “their decoders have multiple layers and they’re dedicated and different for every data instance”, the Transformer hypernetwork of TransINR and IPC predicts 256 latent vectors with 256 dimensionality, the same with our framework. In addition, except for a single instance-specific MLP layer, all the other layers of IPC are not different but shared for every data instance.  If different layers of an INR are dedicated to represent each data instance as the reviewer’s comment, its compression ratio becomes much lower (worse) than our framework as described in our previous response to the reviewer KNfA.
> > > > > > >
> > > > > > >
> > > > > > > **[Connection to LDM]**
> > > > > > > Our overall process for generative modeling in Section 4.4 follows the overall process of latent diffusion models. However, the main difference is the generated latents, while our latents specify the INR of each data instance.
> > > > > > >
> > > > > > >
> > > > > > > **[Novel View Synthesis & Autoencoder]**
> > > > > > > We think that considering our framework of **few-shot** novel view synthesis as autoencoding is a misunderstanding, since a few-shot novel view synthesis is totally different from autoencoding. While autoencoding can reconstruct unseen examples during training, the reconstruction requires full observations of unseen 3D objects to be reconstructed. However, our framework of a few-shot novel view synthesis requires predicting “unseen views” of unseen 3D objects, given only 1-5 views of unseen 3D objects.
> > > > > > > Different from the reviewer’s comment that “only solving novel view synthesis task on single synthetic 3D objects doesn't indicate a big capability boost either.”, a task of few-shot novel view synthesis is non-trivial and challenging task in 3D vision. Note that the recent work, 3DiM (ICLR’23), adopts 1.3B parameters of diffusion model to solve this task on SRN-Cars, since a few-shot novel view synthesis is a challenging task even on a category-specific and synthetic 3D object. The problem setting is also different from the conventional NeRF works to synthesize novel views of a complicated scene, since it requires expensive per-scene training with hundreds of views per each scene. Given only a few support views of unseen 3D objects, synthesizing novel views consistent with the few support views is far from the reconstruction of autoencoding.

---

### Official Review · Reviewer_TnVm · 2023-07-13

**Soundness:** 3 good
**Presentation:** 2 fair
**Contribution:** 2 fair
**Rating:** 5
**Confidence:** 2

**Summary:**

This paper tries to improve the expressive power of neural implicit function modulation by enhancing its ability to localize and capture fine-grained details of data samples. A novel framework for generalizable INR is proposed that combines a transformer encoder with a locality-aware INR decoder. It further utilizes selective token aggregation and the multi-band feature modulation to learn locality aware representation in both spatial and spectral aspects.

**Strengths:**

+ combine locality aware transformer encoder with global neural implicit function to improve the representation quality of local details
+ use selective token aggregation for spatial locality
+ use multi-band feature modulation for spectral locality


**Weaknesses:**

- All the quantitative results are reported with PSNR metric. More evaluation metrics emphasizing local details should be used.
- Limited results reported on novel view synthesis and conditional image generation.
- In conditional generation, the latents are corrupted by Gaussian noise and denoised with a trained diffusion model. If the Gaussian noise is small, the task becomes trivial. An comparison experiment should be done for directly reconstructing with corrupted latents using the implicit neural function.


**Questions:**

Can we interpolate between the latents of two data samples to generate a new sample?

**Limitations:**

Experiment results are limited in evaluation scale and in-depth analysis.

---

> ### Author Rebuttal · Authors · 2023-08-09
>
> **[More Evaluation Metrics to Emphasize Local Details]**
> We compare our framework with IPC [19] to emphasize local details using HF-PNSR-r, which calculates PSNR only using the high frequency components of an image. We have also considered other evaluation metrics such as rFID, SSIM, LPIPS during the rebuttal period. However, for the reviewer’s request to emphasize local details, PSNR is the best metric to measure per-pixel errors, since the others measure the structural or perceptual quality rather than local details. The metrics shown in the table below are PSNRs between the ground truth image and the reconstruction of ImageNette 178x178, after the bottom 5%, 10%, 20%, 40%, and 80% of frequencies are filtered out. The results show that our framework outperforms a previous study, IPC, to reconstruct various ranges of high-frequency details.
>
> |     | remove 5% | remove 10% | remove 20% | remove 40% | remove 80% |
> |:---|:---:|:---:|:---:|:---:|:---:|
> | IPC | 38.81 | 39.18 | 40.35 | 43.37 | 50.20 |
> | Ours | 46.26 | 46.38 | 46.88 | 48.77 | 54.48 |
>
>
> **[Limited Results on Novel View Synthesis and Conditional Image Generation]**
> We emphasize that our experiments on novel view synthesis are not insufficient compared with previous studies. We follow the experimental setting with previous studies such as TransINR [8], and IPC [19], while using the same experimental setting with IPC [19]. Compared with TransINR [8], which has trained their framework only with 1-2 support views, we provide the experimental results with 1-5 support views. While our framework and earlier research have concentrated on category-specific and synthetic datasets, we anticipate that the enhancements in our performance regarding novel view synthesis offer the potential for our framework to extend its learning capabilities to open-domain and large-scale 3D object datasets, such as Shap-E [NewRef: Shap-E], in future endeavors.
>
> For conditional image generation, we have trained a diffusion model on ImageNet 256x256, since ImageNet 256x256 is the renowned and standard benchmark for evaluating class-conditional image generation. Extending the range of conditional generation, such as layout-to-image or text-to-image, would be an interesting future work. However, we consider the in-depth analysis for training diffusion models to generate INRs is beyond the scope of this study.
>
> If the reviewer suggests detailed experiments to add, we will attach the experimental results in the revised version if possible.
>
> [NewRef: Shap-E] Jun, Heewoo, and Alex Nichol. "Shap-e: Generating conditional 3d implicit functions." arXiv preprint arXiv:2305.02463 (2023).
>
>
> **[The Magnitude of Gaussian Noise for Diffusion Model]**
> The task of denoising Gaussian noise is not trivial, since we follow the conventional setting to train diffusion models. Given a diffusion time step $t$, a Gaussian noise $\epsilon$ is added to the localized latents $\mathbf{Z}^{(n)}$ as $\sqrt{\alpha_t} \mathbf{Z}^{(n)} + (1-\alpha_t) \epsilon$, where $\alpha_0=1$ and $\alpha_T=0$. When a diffusion time step is close to $t=0$, the added noise is small. However, when the diffusion time step is close to $t=T$, the latents $\mathbf{Z}^{(n)}$ become a Gaussian noise after noise addition, making the denoising task nontrivial. We note that the generation process of diffusion models starts from pure Gaussian noise to generate INR latents.
>
>
> **[Interpolation of the Latents of Two Data]**
> Although we can interpolate the latents of two samples, the outcomes do not yield semantically meaningful images akin to a straightforward linear interpolation between two images.

---

### Official Review · Reviewer_wXRn · 2023-07-17

**Soundness:** 3 good
**Presentation:** 3 good
**Contribution:** 3 good
**Rating:** 5
**Confidence:** 3

**Summary:**

This paper aims to enhance the performance of generalizable implicit neural representation locality-aware model designs. A transformer encoder is applied to convert image patches into latent tokens, which the proposed locality-aware decoder composed of the selective token aggregation and the multi-band feature modulation is based on to predict outputs. Experiments on tasks including image reconstruction and novel view synthesis are performed to demonstrate the proposed method's state-of-the-art performance.

**Strengths:**

+ The proposed locality-aware decoder containing selective token aggregation and multi-band feature modulation is novel and effective.
+ Extensive experiments show that the proposed method achieves good performance.
+ The paper is clear and easy to follow.

**Weaknesses:**

- It is not very straightforward to get the idea of  "Locality" in this work. Any empirical evidence like visualization in Selective Token Aggregation to reveal how the locality is enhanced?
- Analysis and comparison of model/runtime efficiency are missing.
- Some important implementation details are missing, e.g. what's the value of L and what's the impact of this hyperparameter?

**Questions:**

see weakness.

**Limitations:**

- No analysis for failure cases.
- Qualitative results in the paper are limited, and should be more diverse.

---

> ### Author Rebuttal · Authors · 2023-08-09
>
> **[The Idea of Locality]**
> In our study, “locality” describes a concept that the features in a data instance have potentially a high correlation with each other, where the distance between their corresponding coordinates is close. For example, in a 2D image, pixels located close to each other tend to have similar RGB colors when the two pixels are located in a nearby position. Thus, the notion of “locality-aware" implies that our latent tokens learn to contain the information of a local region for efficient and effective representation of a data instance.
>
> Figure 5 provides the empirical evidence of enhanced locality-awareness, since the previous study, IPC, has overlooked modeling the local information of a data instance. Figure 5 shows that each latent of IPC affects most coordinates' features, failing to model local information. However, our framework shows that each latent covers a certain region of the local area, enhancing the locality-awareness of our framework.
>
>
> **[Analysis and Comparison of Model Efficiency]**
> For the experiments on ImageNette 178x178 in Figure 1, our framework has 0.9% more parameters, 44.14M, than IPC having 43.75M of trainable parameters. The runtime of IPC is 78 seconds per  training epoch, but our framework takes 90 seconds. However, Figure 1 shows that our framework is significantly more efficient and effective than IPC and TransINR, despite a 15% longer runtime per training epoch.
>
>
> **[The Value of $L$]**
> Our experiments use $L=2$ as described in Line 206 and Line 238, since $L$ is the number of frequency bandwidths. We will explicitly specify the total number of $L$ in our revised version. The experiments to study the impact of the hyperparameter $L$ are attached to Appendix B.1 in our supplementary material.
>
>
> **[Analysis for Failure Cases]**
> Although our framework significantly improves the performance of a generalizable INR, the reconstruction performance on 1024x1024 image resolution is still incapable of perfectly reconstructing all high-frequency details in original images. In addition, the qualitative results on novel view synthesis show blurry examples due to the lack of the training objective of generative modeling to synthesize unseen views. We will attach the explanation of failure cases above to inform the research community about the boundary of this study.
>
> **[Diverse Qualitative Results]**
> Please refer to the attached supplementary material for more diverse qualitative results. Our supplementary material includes various examples of novel view synthesis with different numbers of support views, image reconstruction with 256x256, 512x512, and 1024x1024 resolutions, class-conditional image generation on ImageNet, and additional visualization for locality analysis.

---

> > ### Comment · Reviewer_wXRn · 2023-08-22
> > **Acknowledgement**
> >
> > Thanks for the response and it resolved some of my concerns. I would like to keep my rating.

---

### Official Review · Reviewer_AhKU · 2023-07-23

**Soundness:** 2 fair
**Presentation:** 1 poor
**Contribution:** 3 good
**Rating:** 4
**Confidence:** 4

**Summary:**

The paper focuses on the task of training a single coordinate-based neural network to represent multiple scenes or instances. There are two main technical contributions that improve the quality of these generalized representations: (1) a Transformer-based encoder that extracts localized features of each target instance (e.g. image or scene), and adaptively weights these localized features during inference at different coordinate locations, (2) the coordinate-based neural network operates coarse-to-fine in the frequency domain, taking in high-frequency instance-specific features at earlier layers and lower-frequency instance-specific features at later layers, so that higher-frequency features are processed by a deeper network. The paper includes compelling results on medium and high resolution image datasets, and promising preliminary results on few-shot novel view synthesis. Ablation studies show that of the two technical contributions, neither alone yields improvement but both together do, compared to the main baseline Instance Pattern Composers.

**Strengths:**

These two technical ideas make sense, particularly the idea of having localized features, and the results on image datasets are very compelling—though some important omitted details make these difficult to fully interpret, as does my own lack of familiarity with the baseline methods. Ablation studies and more in-depth analysis of learned features (figure 5) are also interesting.

**Weaknesses:**

The paper writing/presentation could be substantially improved. The first roughly 2 pages of the paper are laden with jargon (and some typos/grammatical issues), making the actual new ideas in the paper difficult to tease out. As a reader who has worked in implicit neural representations but not generalizable ones, many parts of the paper required some assumptions or terms that might be common to those who work in generalizable INRs but unfamiliar to researchers in even a very adjacent area. Some examples: what exactly is a latent (ie what is the input used to produce a latent)? What is modulation (in signal processing this would be element-wise multiplication)? The methods section does largely (though not fully) clarify the method and the new ideas, but leaves me wondering about the motivation behind some of the design decisions that are stated without much explanation. I list these remaining questions in the “questions” section of the review, along with some questions about the experimental setup of the results.

One separate comment/suggestion is about the clarity of the figures. Figure 1 shows that the proposed method trains a lot faster/better than two baselines, but doesn’t explain what the task is (just the dataset). Figure 2 gives a helpful overview, but doesn’t actually explain the two core new ideas of the paper. For example, from the figure and caption I can’t tell what is different about the two yellow blocks (one is for high-frequency features and one for low-frequency features, but I didn’t find this out until I read 2 pages past the figure). Some terms in the figure are also not defined; for example I assume that “FC” means fully-connected, but I’m not sure how this differs from “Linear”. Figure 3 shows a compelling comparison to prior work, but I would encourage the authors to include the ground truth image for full comparison.


**Questions:**

- I am still not fully sure what exactly is the input to the model during inference. Is it basically an autoencoder, taking an image as input and compressing it into latents and then decoding them into the image again? Likewise for the generative experiment (figure 6), how are these latents generated?
- Equation 2 describes the Fourier featurization; it appears to use the same featurization as the original NeRF paper (with axis-aligned frequencies) rather than e.g. a random Gaussian (not axis-aligned) set of frequency vectors as in the “Fourier Features Let Networks Learn…” paper, though both of these papers are cited. I wonder why the authors chose to use the original Fourier featurization rather than the newer version?
- Equation 5 describes multi-head attention as the mechanism for aggregating/weighting the localized latent features when performing inference on a given coordinate vector. I wonder if the authors considered a more straightforward kernel function (e.g. Gaussian kernel based on Euclidean distance, or even a learned kernel) or what the motivation was for using multi-head attention.
- Equations 6 and 7 describe a process for separating the modulation vector into different frequency bands, by using a shallow MLP with inputs based on different Fourier features (in the desired frequency band). I wonder why the MLP is necessary here, compared to just taking the subset of Fourier features that are in each frequency band?
- Tables 1 and 2 show compelling numerical comparisons vs Learned Init, TransINR, and IPC. The accompanying text notes that the capacity of the encoder, latent tokens, and decoder are matched among all methods, except for the modulation methods. So I wonder how the capacity of the modulation methods compares?
- I also wonder what the total training time is on each dataset, compared to prior work.
- Figure 7 shows that test-time optimization of all parameters outperforms per-sample optimization. How is this possible? I wonder if training longer would allow the per-sample FFNet to “catch up”, or if the FFNet here has lower capacity, or if there is some other explanation for how a generalized model could outperform a more specialized model. Maybe the generalized model is starting from a better initialization and thus reaches a better optimum?

**Limitations:**

Limitations are not really discussed, beyond mentioning some directions for future work--but for this topic I believe this is appropriate. I do not foresee direct negative societal impact from this work, though like all image processing research it has the potential to be misused e.g. towards surveillance or other harmful ends.

---

> ### Author Rebuttal · Authors · 2023-08-09
>
> **[Improvement of Introduction]**
> As per suggestion, we will revise our manuscript to clarify the terminologies in the Introduction for better understanding as follows. The latents refer to the outputs of our Transformer Encoder, corresponding to the positions of learnable input tokens. The input of the Transformer Encoder is the concatenation of data tokens and learnable tokens, where the number of learnable tokens equals the number of latent tokens.
> Weight/feature modulation refers to the modification of a weight/feature by element-wise multiplication or addition to adapt a shared weight/feature for a data instance. Thus, the Transformer encoder extracts a set of latent tokens to modulate the shared parts of generalizable INRs to represent a data instance.
> Our study is primarily motivated by the limitation of SOTA methods, TransINR and IPC, which do not consider the locality in the data. We have elaborated on why the previous approach, IPC, cannot consider the local information of data in Lines 134-138 in Section 3.3.1. We will include the details of previous studies to clarify the motivation of our study and the limitations of previous studies.
>
>
> **[Typos and Grammatical Issues]**
> We will fix all typos and grammatical issues.
>
>
>
> **[Suggestions for the Clarity of Figures]**
> Reflecting the reviewer’s suggestions, we will add and clarify our figures as follows:
> - Figure 1: The task is image reconstruction.
> - Figure 2: The two frequency features have different bandwidths. We will replace “FC” with “Linear.”
> - Figure 3: We will add the ground truth image for detailed comparison.
>
>
> **[Inputs to the model]**
> Regarding explaining the input of Transformer Encoder, please refer to **[The Inputs of Transformer Encoder]** in our responses to Reviewer QDhX. Our framework for image reconstruction can be viewed as an autoencoder. However, our framework for synthesizing novel views should not be seen as an autoencoder, given that it involves the synthesis of previously unseen perspectives.
>
> We adopt the two-stage framework to generate an image for conditional image generation. We first train our framework on ImageNet 256x256 to represent an image as a set of localized latents for INRs. After representing each image as localized latents, a diffusion model is trained following the experimental setting in Appendix A.3. After the training, the diffusion model gradually denoises the corrupted latents, which start from isotropic Gaussian noises, to generate new images.
>
>
>
> **[Fourier Featurization]**
> We adopt the Fourier featurization with axis-aligned frequencies in the original NeRF paper to ensure the stable and high performance of generalizable INRs. Since the random initialization of Fourier features [36] requires a careful selection of the variance for each sample, adopting random Fourier features deteriorates the performance of generalizable INRs, as shown below. We also emphasize that recent NeRF models [NewRef: RefNerf] and generalizable INRs [8, 10] have still adopted the Fourier featurization with axis-aligned frequencies.
>
> | ImageNette 178x178 | PSNR |
> |:-----|:-----:|
> | IPC | 34.11 |
> | Ours | 37.46 |
> | IPC w/ random FF | 29.27 |
> | Ours w/ random FF | 30.94 |
>
> [NewRef: Ref-Nerf] Verbin, Dor, et al. "Ref-nerf: Structured view-dependent appearance for neural radiance fields." 2022 IEEE/CVF Conference on Computer Vision and Pattern Recognition (CVPR). IEEE, 2022.
>
> **[Multi-Head Attention in Selective Token Aggregation]**
> We exploit multi-head cross attentions for selective token aggregation, since using cross-attention is an intuitive choice within the context of modern deep learning architectures. As learned kernel functions, cross-attentions can consider various similarity patterns between projected queries and keys, instead of a tailored similarity pattern. As shown below, adopting multi-head cross attentions improve the reconstruction performance on FFHQ 178x178, compared with single-head cross-attention, which is close to a learned kernel function.
>
> | | ImageNette 178x178 | FFHQ 256x256 |
> |:------|:-----:|:-----:|
> | Ours (2 heads) | 38.72 | 39.88 |
> | w/ single-head | 37.46 | 38.01 |
>
> **[A Linear Layer in Multi-Band Feature Modulation]**
> We add a Linear layer in Equations (6) and (7) to exploit complex frequency patterns, improving the performance. While the Fourier features consist of periodic patterns along an axis, the frequency patterns in Equation (6) can also include non-periodic patterns. Note that IPC [19] also uses a similar design, while modulating the second MLP layer to exploit complex frequency patterns. The linear layer in Equation (7) is used to process the modulation vector according to each frequency bandwidth, motivated by the design of separate projections for (query, key, value) in self-attention. The results below also show that removing the linear layers in Equations (6) and (7) significantly deteriorates the image reconstruction performance on ImageNette 178x178.
>
> | | ImageNette 178x178 |
> |:----|:----:|
> | Ours | 37.46 |
> | w/o Linear in Eq. (6) | 31.95 |
> | w/o Liner in Eq. (7) | 32.07 |
> | w/o Liner in Eq. (6) and (7) | 31.57 |
>
>
> **[The Capacity of the Modulation Methods]**
> The modulation capacity is determined by the size of latents to represent instance-specific information, except for the shared weights. TransINR, IPC, and our framework commonly use latents of size 256$\times$256 as instance-specific information to modulate a shared architecture.
>
> **[Total Training Time compared to Prior Work]**
> Please refer to our responses to Reviewer wXRn.
>
>
> **[Performance of Test-Time Optimization]**
> While per-sample optimization of FFNet starts from a random initialization, our framework provides a good initialization of an INR having high performance, as shown in Figure 7. Since our test-time optimization also updates whole parameters of the INR, it is reasonable to expect that improved initialization could lead to savings in training costs.

---

> > ### Comment · Reviewer_AhKU · 2023-08-13
> >
> > The rebuttal addresses many of my concerns, but I still find the explanations in the paper to be needlessly confusing--even the first paragraph of the rebuttal does not really explain its terms (or rather, it explains terms by introducing more terms, which doesn't really help). I'm still on the fence about this paper because although the results are impressive quantitatively and the high-level ideas make sense, I'm not sure that the explanations are sufficient for other researchers to really understand what is going on (at least not without putting in a lot more effort as a reader than is common) or to build productively on the work. I won't stand in the way of acceptance given the strength of the results, but I do strongly encourage the authors to make sure that the writing/presentation in the final version is clear to readers in very adjacent (e.g. single-scene representation) if not identical research areas, to maximize the potential impact of the paper.

---

> > > ### Author Response · Authors · 2023-08-14
> > >
> > > Dear Reviewer AhKU,
> > >
> > > To avoid any confusion, we will revise our definitions by citing earlier works that provide more detailed definitions, including modulations [NewRef: Film] and the latents [NewRef: Perceiver], since these terms are frequently used in subsequent works without detailed definitions.
> > >
> > >  [NewRef: Film] Perez, Ethan, et al. "Film: Visual reasoning with a general conditioning layer." Proceedings of the AAAI conference on artificial intelligence. Vol. 32. No. 1. 2018.
> > >
> > >  [NewRef: Perceiver] Jaegle, Andrew, et al. "Perceiver: General perception with iterative attention." International conference on machine learning. PMLR, 2021.

---

### Official Review · Reviewer_QDhX · 2023-07-26

**Soundness:** 3 good
**Presentation:** 3 good
**Contribution:** 3 good
**Rating:** 6
**Confidence:** 3

**Summary:**

The paper tackles the important problem of bridging the gap between generalizable implicit neural representations (INR) and per-sampled trained ones. The core hypothesis is that previous generalizable INRs failed to capture local details in the global latent code due to their inductive bias. The idea of the proposed method is to equip learning such INRs with a Transformer encoder, which selects and encodes local information into multiple tokens. The paper elaborates on all parts of the pipeline and presents the empirical study on image reconstruction and novel view synthesis with shapenet. Overall, the proposed method shows a great advantage over the prior art. Ablation studies are thorough.

**Strengths:**

The idea behind the method is novel and interesting. The paper borrows the best ideas from other domains, including transformers and nerf. Experimental studies on FFHQ and Shapenet are convincing. The paper is well-written and rather polished.

**Weaknesses:**

The introduction is a bit too broad and needs more specificities about the proposed method. The statements from the title, abstract, and intro, set expectations for something more generic than what can be processed by a transformer encoder.

The frequency decomposition mechanism is not very well explained. Particularly, the choice of two blocks for frequency decomposition is never explained nor ablated. Would the performance keep growing if more of those blocks had a different frequency coverage scheme?

The locality property of the produced tokens is also not very clear. On the one hand, the authors claim that the tokens are locality aware; on the other that thanks to permutation equivariance of the attention mechanism in the transformer encoder, the tokens form an unordered set. It would be better to explain these aspects more carefully (including in the rebuttal).

List of typos
- L49: outperformance

**Questions:**

In Fig. 2, how do the transformer inputs form in each considered setting (image / NVS)?
Does the framework require retraining everything from scratch for every new resolution?
Can any of the pretrained backbones be used?
Can the authors think of a way to visualize the effect they discuss regarding ordering frequencies based on the depth of the layer in the decoder?
What do the results look like for the best possible setting for 1Kx1K resolution in Table 2?

**Limitations:**

Discussed adequately in the conclusion. However, it would be interesting to learn about dealing with arbitrary resolutions, or other forms of inputs not directly amenable to transformer encoders.

---

> ### Author Rebuttal · Authors · 2023-08-09
>
> **[Broad Introduction]**
> We will revise the Introduction by adding the following specific description of our transformer encoder. Specifically, we will add the detailed explanations about the use of cross-attention in selective token aggregation to Lines 43-44. The cross-attention is used for each coordinate to extract the information of the latents, which are the output of the Transformer Encoder. We will also describe how multi-band frequency decomposition is designed in Lines 45-46.
>
>
> **[Choice of the Number of Blocks for Frequency Decomposition]**
> We will revise the explanations for better understanding of the frequency decomposition mechanism. In addition, Appendix B.1 includes the ablation study, which the reviewer has mentioned, on the number of blocks $L$ for frequency decomposition. For image reconstruction of FFHQ 256$\times$256, 512$\times$512, 1024$\times$1024, the performance improves with respect to $L$ increasing from 1 to 3, but the performance saturates at $L \geq 3$. In our main experiments, we use $L=2$, considering the trade-off between computational costs and performance.
>
>
> **[Clarifying Locality Property of the Produced Tokens]**
> Figure 5 demonstrates the locality property of the produced tokens. Figure 5 visualizes that each latent token captures the local information of a data instance and affects the pixels/rays in a certain local area.
>
> We will clarify how the permutation-equivariance of self-attention affects the design of our framework in terms of locality property as follows. The permutation-equivariant of self-attention in the Transformer encoder makes our framework not assume the local structures of data and latent tokens. That is, we do not assume the permutation of the location of latent tokens (ordering of latent tokens), but consider the latent tokens as a set of local information. During training, each latent token learns to capture the local information of data, while covering whole regions to represent a data instance. This property enables our framework to be readily applied to diverse data with non-grid coordinates. For example, determining the order and size of local regions is not straightforward for Plücker coordinates of rays. However, self-attention enables each local latent to capture the local information of data, while the set of local latents represents the information of whole rays.
>
> **[Typos]**
> Thanks for the detailed comment. We will fix all typos in our manuscript.
>
> **[The Inputs of Transformer Encoder]**
> A transformer input is the concatenation of image patches and learnable tokens, as described in Appendix A.1 and A.2. For image reconstruction, an image is represented as a set of patches, where each patch has $P \times P$ size. We use P=9, 16, 32, 48 for 178$\times$178, 256$\times$256, 512$\times$512, and 1024$\times$1024 resolution, respectively. For 178$\times$178 and 1024$\times$1024 resolution, images are zero-padded to make the size evenly divisible by the patch size. We use zero-padding of 1 and 16 pixels on every side, respectively.
>
> For novel view synthesis, we use the Plücker coordinate to represent the information of the rays in the rendering view of a 3D object. Given rendering images of support views, we concatenate the ray coordinates with pixels along the channel, and then patchify the support views using $P=8$ patch size. Then, we concatenate the patches of all support views with learnable tokens for the input of our Transformer.
>
> **[Retraining for new resolution]**
> Yes, we train our framework for each resolution of the dataset.
>
> **[The Use of Pretrained Backbones]**
> Since conventional pretrained models are not trained for the latents of INRs, we believe that such pretrained backbones cannot be utilized within our framework.
>
> **[Visualization of the Effect regarding Frequency Ordering]**
> In the additional pdf file for author responses, we visualize the effect of frequency ordering on the reconstruction of high-frequency details. We visualize the pixel-wise reconstruction error for models trained on ImageNette 178x178 with $(\sigma_1, \sigma_2)=(128, 32)$ and $(\sigma_1, \sigma_2)=(32, 128)$. Our design choice $(\sigma_1, \sigma_2)=(128, 32)$ shows superior performance in reconstructing high-frequency details of data.
>
> **[Qualitative result in 1024x1024 resolution]**
> Appendix B.3 includes the qualitative results for 1024x1024 image reconstruction.
>
> **[Limitations]**
> Thanks for the constructive comments. We agree that extending our framework to support arbitrary resolution will be an interesting future work. We will also describe the limitation of our framework that requires an amenable form of input for transformer encoders.

---

> > ### Comment · Reviewer_QDhX · 2023-08-20
> > **Acknowledgement**
> >
> > I thank the authors for carefully responding to my questions and concerns. I also checked the other reviews and ongoing conversations, especially in this thread https://openreview.net/forum?id=XqcXf7ix5q&noteId=mtClW0NJlm , and I could find resonating and well-justified arguments on both sides, the authors, and Reviewer KNfA. One thing is clear - works in our domain are getting harder to disseminate due to the lack of rigorous and established processes for many aspects of scientific writing. In this case - whether INR is more correctly defined as "baking the entire instance in the weights" (Reviewer's KNfA point of view) or "regardless of the size of the instance-specific code, it is an INR, because it expects an extra coordinates input" (the authors), is a matter of naming conventions.
> >
> > In my view, the "I" in INR is primarily borrowed from the notion of implicit functions, which were used as a representation for SDFs, encoding surfaces implicitly. The implicity in question stems from the fact that one has to actually solve for isosurfaces of F(x) = 0. NeRFs are also implicit due to the ray marching color accumulation process.
> >
> > It would be good if the authors of the paper took some time to carefully work through all the terms and notation, and explained with references what makes their representation implicit, what the overall field of INR properties consists of, and how generalization fits this landscape at all. Currently, generalizable INRs only fit the definition proposed by the authors and rather do not fit the other two; this should be reconciled.
> >
> > Considering the paper's empirical contribution and discounting for the naming conventions, I would like to keep my score, conditioned on the authors properly addressing writing and maybe setting some notation for the subsequent works.

---

> > > ### Author Response · Authors · 2023-08-21
> > >
> > > We appreciate the reviewer’s discussion and feedback. In particular, we fully agree with your emphasis on “rigorous and established processes for scientific writing,” and feel that these discussions with reviewers greatly help in that sense indeed. We are confident that our final camera-ready can address all such presentation issues.
> > >
> > >
> > > We agree with the reviewer’s view that the "I" in INR is primarily borrowed from the notion of implicit function formulation, while we have focused more on the concept of baking an instance in neural network weights in discussion with reviewer KNfA. One of early INRs, SIREN (NeurIPS’20), also presents that the “I” of INRs stem from solving implicit problem formulation of $F(\mathbf{x}, \Phi, \nabla_\mathbf{x} \Phi, \nabla^2_\mathbf{x} \Phi, …)=0$, where $\Phi: \mathbf{x} \mapsto \Phi(\mathbf{x})$ and $\mathbf{x}$ is a coordinate, and casts the optimization of implicit problem formulation into a loss function in Eq. (3) of the paper. We believe the formulation in the SIREN paper would be a good starting point for us to reconcile different views and applications. Solving for the isosurfaces of SDFs, which the reviewer mentioned, can be described as shown in Eq. (6) and Section 4.2 of the paper and our image reconstruction can also be formulated under the same setting of implicit problem as shown in Section 3.1 of the paper, although it is simpler than SDFs. Light fields, which replace a volume rendering process of NeRFs with directly predicting the rendering results, also preserve the form of implicit problem, while simplifying the optimization problem of NeRF as a simple first-order optimization problem [Ref-1].
> > >
> > >
> > > Note that what we wanted to emphasize in discussion with the reviewer kNfA is that the notion of “implicit” does not restrict the scope of instance-specific parameterization. Our view is not different from that of the implicit problem formulation above; generalizable INR follows the formulation of implicit problem, while a function $\Phi$ is conditioned as $\Phi(\cdot | \mathbf{Z})$, where $\mathbf{Z}$ is the latents of each data instance. Since $\Phi$ adopts a parameterized coordinate-based neural network, the implicit representation of data corresponds to the (part of) parameters of coordinate-based neural networks under the implicit formulation.
> > > As the reviewer requested, we will do our best to revise our paper with clear notions and rigorous formulations for INRs based on the extensive discussions with the reviewers.
> > >
> > >
> > > [Ref-1] Sitzmann, Vincent, et al. "Light field networks: Neural scene representations with single-evaluation rendering." Advances in Neural Information Processing Systems 34 (2021): 19313-19325.

---

### Author Rebuttal · Authors · 2023-08-09

We appreciate all reviewers's constructive comments to improve our paper.
We have tried our best to sincerely respond to all concerns and questions.

---

### Decision · Program_Chairs · 2023-09-21

**Decision:**

Accept (poster)

**Comment:**

This paper received a mixing of ratings with strengths on the novelty of idea, good performance on multiple benchmarks, and weakness on the unclear descriptions (Reviewer AhKU) and definition of generalizable INRs (Reviewer KNfA).  AC has read the reviewers’ feedbacks, the authors’ rebuttals, and the paper. For the concern from Reviewer KNfA on the definition of generalizable INR,  AC agrees with Reviewer QDhX that representing the signal instance with model weights and using latent features can both be considered as INR and thus discounts this factor in making decision.

As suggested by Reviewer AhKU, the authors need carefully revise the paper to properly addressing writing problem.

The frequency decomposition mechanism is not very well explained (Reivewer QDhX). The authors have visualized the effect of frequency ordering on the reconstruction of high-frequency details. However, there is still a lack of deep analysis. For example, why the design choice (128,32) shows superior performance? whether a deeper MLP path corresponding the learning of higher frequency features and why (how to demonstrate that)? In addition, the cross-attention for selective token aggregation shares some similar idea with the work “attentive neural processes, ICLR19”. The authors could add the discussion.

Given the strength of the performance and interestingness of the idea, AC reached a decision to acceptance.
AC strongly encourages the authors to revise the paper to make sure that the writing/presentation in the final version is clear to readers and add deep analysis on whether features align the expectation and whether the content features align with the frequencies of the position encoding.